# Neural-Inspired Modeling of Auditory Selection and Compensation for Audio-Visual Speech Separation

Xinmeng Xu [1]   Haoran Xie [1]   Xiaohui Tao [2]   Lin Li [3]   S. Joe Qin [1]

## Abstract

Current audio-visual speech separation (AVSS) models typically rely on implicit multimodal fusion, but the absence of explicit modality alignment and reliability modeling often causes semantic misalignment and contaminates speech representations. The brain addresses this with a hierarchy: top-down auditory selection uses visual priors to maintain target-consistent acoustics, while bottom-up cross-modal compensation integrates temporally aligned articulatory cues to reconstruct and stabilize speech. Guided by this principle, we present Neuro-SCNet, an AVSS architecture that makes selection and compensation explicit and reliability-aware. The Auditory Selection Mechanism applies top-down, visually guided gain along the audio pathway to isolate target time-frequency units and suppress distractors. The module preserves the auditory trace with an identity bypass and adds controlled visual refinements via a residual path. A synchrony-driven gate reduces the influence of low-confidence visual cues. Additionally, a lightweight pre-alignment for visual feature pre-processing estimates and corrects small temporal offsets, and a compact magnitude-phase encoder is used to preserve fine acoustic detail to stabilize reconstruction. Evaluations on LRS2, LRS3, and VoxCeleb2 show state-of-the-art separation with improved efficiency, supporting the value of explicit selection and reliability-aware compensation.

[1]Division of Artificial Intelligence, Lingnan University, Tuen Mun, Hong Kong SAR [2]School of Mathematics, Physics and Computing, University of Southern Queensland, Toowoomba, Australia [3]School of Computer Science and Artificial Intelligence, Wuhan University of Technology, Wuhan, China. Correspondence to: Haoran Xie (Corresponding Author) <hrxie@ln.edu.hk>.

*Proceedings of the $43^{rd}$ International Conference on Machine Learning*, Seoul, South Korea. PMLR 306, 2026. Copyright 2026 by the author(s).

## 1. Introduction

The cocktail–party problem captures the human capacity to attend to one speaker amid competing talkers and background noise (Cherry, 1953; Arons, 1992). Audio-only speech separation (AOSS) seeks to model this ability (Jiang et al., 2025; Shin et al., 2024; Wang et al., 2023), yet its accuracy drops in unconstrained mixtures, open-set numbers of speakers, heavy temporal overlap, reverberation, and nonstationary noise, especially when the number of talkers departs from training regimes (Yu et al., 2017; Hershey et al., 2016; Xu et al., 2020). Neuroscience shows that visual articulatory cues (lip motion, facial dynamics) provide complementary information that facilitates speech discrimination under such conditions (Li et al., 2018; Stenzel et al., 2019). Motivated by this evidence, audio-visual speech separation (AVSS) integrates time-synchronized visual input with acoustic features and has achieved clear gains over AOSS across diverse settings (Ephrat et al., 2018; Gabbay et al., 2018; Afouras et al., 2018b).

Recent AVSS systems place cross-modal attention inside the encoder so that the audio and visual streams can exchange information (Xu et al., 2025a; Li et al., 2024b; Xu et al., 2023). AV-CrossNet implements early fusion with complex spectral mapping in the time–frequency domain (Kalkhorani et al., 2025), whereas IIA-Net uses hierarchical intra- and inter-modality attention to gate one stream with the other (Li et al., 2024b). These choices improve semantic alignment by reweighting features, yet the reconstruction goal remains twofold: the model must infer the correct semantics and, at the same time, keep fine acoustic detail (Gerkmann et al., 2015; Yao et al., 2025). Visual input helps the first part by providing articulatory cues for phonetic disambiguation, but it contributes little spectral detail, especially at low frequencies (Afouras et al., 2018c). Meanwhile, the audio signal mixes target speech with interference; when shared layers process both streams, the interference can drive misleading gradients and pull the representation away from the target (Baltruaitis et al., 2019). Temporal misalignment then makes this problem harder by tying visual priors to nonmatching acoustic events, which can amplify distractors instead of suppressing them.

A neuroscience-grounded two-stage mechanism addresses

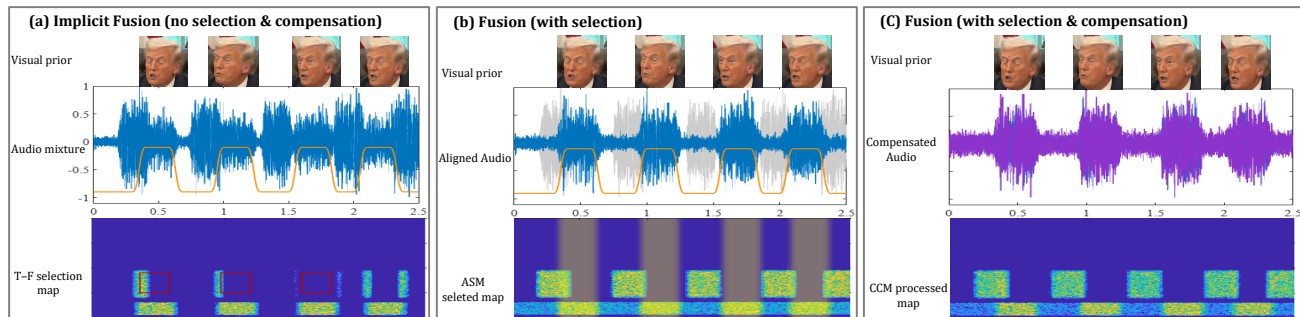

*Figure 1.* The idea of our work, contrasting implicit fusion with explicit selection and compensation. Blue denotes the audio mixture (and the bypass in c), orange the visual prior, and purple the compensated output; the bottom-row spectrograms are magnitude displays in dB (generating procedure see **Appendix Sec.E**). (a) Implicit fusion: a time-only mask guided by the visual prior ends up lifting distractor bands; red boxes mark these coupled distractors. (b) Selection (ASM): the selection mask preserves target-consistent time and frequency regions and suppresses others. (c) Compensation (CCM): an identity bypass with a reliability-weighted visual residual restores structure without boosting distractors.

these issues (Li et al., 2018; Ghazanfar & Schroeder, 2006). First, auditory selective attention applies early, top-down gain control guided by visual and prefrontal signals in auditory cortex, isolating target-consistent regions of the time–frequency representation and suppressing interference (Mesgarani & Chang, 2012; Lakatos et al., 2013; Golumbic et al., 2013; Schroeder & Lakatos, 2009; Beauchamp et al., 2004b). By gating non-target activity before deeper processing, this stage limits the propagation of misleading updates through downstream circuits. Second, cross-modal compensation arises from bottom-up integration in multisensory areas such as the superior temporal sulcus, where temporally aligned articulatory cues are combined with incomplete auditory traces to infer missing phonetic content and stabilize the percept (Beauchamp et al., 2004a; Van Wassenhove et al., 2005). This hierarchy filters interference at the outset and then restores degraded information with complementary visual evidence, motivating a clear separation between selection and compensation in computational models of AVSS.

Guided by this hierarchy, we introduce Neuro-SCNet[1], a two-stage AVSS model that makes selection and compensation explicit. The Auditory Selection Module applies visually guided gain before fusion to pass target-consistent time-frequency regions and suppress distractors, which also limits updates from non-target areas. The Cross-Modal Compensation Module then fuses the attended audio with articulatory cues and uses residual correction with feedback to recover missing content; an identity bypass preserves the attended trace, and a reliability controller down-weights uncertain visual evidence. Figure 1 links problem and solution: (a) without selection, fusion that follows only the visual timeline lets non-target bands rise with the target; (b) selection applies visually guided gain before fusion to pass only target-consistent regions; (c) compensation adds

a reliability-weighted visual residual on top of the identity path so the base is kept and visual cues contribute only when alignment is confident.

To preserve fine acoustic detail while modeling cross-modal semantics, we use a compact dual-path encoder-decoder: one path extracts semantics and the other carries magnitude and phase. A lightweight visual pre-processing step estimates a small offset to improve temporal robustness. We evaluate on LRS2, LRS3, and VoxCeleb2. Compared with AV-CrossNet (Kalkhorani et al., 2025), Neuro-SCNet improves SI-SNRi by 2.34%, SI-SDRi by 4.03%, and PESQ by 0.84% on average, with 6.3M parameters and 17.8G MACs. Our contributions are summarized as follows:

- A brain-inspired AVSS framework that separates selection and compensation, enabling explicit suppression of interference and targeted restoration of missing content.

- A modular design: ASM filters and reorganizes audio features under visual semantic guidance; CCM completes degraded speech representations via aligned visual cues and residual feedback.

- A compact dual-path encoder-decoder that separates semantics from magnitude and phase, preserving intelligibility and fidelity at the same compute. A visual pre-processing step estimates a bounded-window offset and applies a differentiable shifter to improve temporal robustness.

## 2. Related Works

**Audio–Visual Speech Separation.** AVSS uses visual cues such as lip motion to improve separation under noise and overlap. Early dual-encoder models processed audio and video independently and fused once by concatenation or at-

---

[1] github.com/XinmengXu/NSC-Net

tention over high-level embeddings (Ephrat et al., 2018; Gabbay et al., 2018; Afouras et al., 2018b; Lee et al., 2021). Later work added cross-modal attention to weight one stream by its temporal match to the other (Xu et al., 2025a; 2023; Makishima et al., 2021), and combined intra- with inter-modal attention to clean each stream before fusion (Li et al., 2024b). Multi-stage and multi-scale designs further refined features across iterations and resolutions (Xu et al., 2025a; Li et al., 2024a; Xu et al., 2022). Despite these advances, most systems keep target selection and content restoration entangled and rely on implicit alignment, which limits control and robustness when one modality is degraded (Lee et al., 2021; Hu et al., 2023).

**Neuro-Inspired Mechanisms in Multimodal Perception.** Neuroscience indicates two complementary processes for robust speech perception: auditory selective attention and cross-modal compensation (Ghazanfar & Schroeder, 2006; Li et al., 2018; Peelle & Sommers, 2015). Prior AVSS studies approximate the first by suppressing non-target energy or gating audio with visual cues (Lee et al., 2021; Wu et al., 2022; Tao et al., 2025; Shinn-Cunningham et al., 2015), consistent with top-down modulation in auditory cortex (Mesgarani & Chang, 2012; Lakatos et al., 2013; Golumbic et al., 2013). The second process, adding aligned visual evidence to complete missing auditory content, has mostly appeared implicitly in recurrent or layered fusion (Xu et al., 2025a; 2022; Iuzzolino & Koishida, 2020) without an explicit functional separation or guarantee. This work closes that gap by separating selection from compensation and by adopting constructs that map to these processes directly, enabling explicit control and measurable behavior.

## 3. Method

### 3.1. Neuroscience-Grounded Theory

This subsection adapts two neuroscience mechanisms into simple modeling rules and states the basic claims that support their use. We model selective attention as early gain control on time–frequency features, and multi-modal compensation as predictive coding that adds visual structure without replacing the auditory pathway. Each rule is accompanied by a short equation and a brief biological note.

Selective attention in auditory cortex amplifies target-consistent responses and suppresses distractors before deeper processing, with modulation from visual and prefrontal inputs (Mesgarani & Chang, 2012; Lakatos et al., 2013; Golumbic et al., 2013; Schroeder & Lakatos, 2009; Beauchamp et al., 2004b). A simple population-level abstraction is multiplicative gain. Let $A$ denote an auditory representation and let a gate $m \in [0,1]^{T \times F}$ produce the selected signal $A_{\text{sel}} = m \odot A$. For any loss $\mathcal{L}$, the gradient

satisfies

$$\frac{\partial \mathcal{L}}{\partial A} = m \odot \frac{\partial \mathcal{L}}{\partial A_{\text{sel}}}, \qquad (1)$$

so units attenuated in the forward pass are proportionally attenuated during backpropagation, which limits distractor-driven updates of shared parameters. Task-directed gating can be posed as an information-bottleneck objective

$$\max_{m, \psi} \mathbb{E}[\log q_{\psi}(S \mid A_{\text{sel}})] - \beta \, \mathcal{C}(m), \qquad \beta > 0, \quad (2)$$

where the first term lower bounds $I(S; A_{\text{sel}})$ and $\mathcal{C}(m)$ penalizes capacity, for example through sparsity or rate. This formulation accords with neurophysiological evidence that attention induces approximately multiplicative gain changes in spectrotemporal responses (Mesgarani & Chang, 2012; Lakatos et al., 2013; Golumbic et al., 2013; Schroeder & Lakatos, 2009; Beauchamp et al., 2004b). It explains why early gating increases target throughput and reduces distractor-driven plasticity. When audio-visual synchrony is weak or ambiguous, the visual drive shaping $m$ is attenuated by design, and the statements above remain valid as the gate defaults toward neutral weighting.

Multisensory regions such as superior temporal sulcus combine temporally aligned articulatory evidence with incomplete auditory traces and iteratively reduce prediction error (Beauchamp et al., 2004a; Van Wassenhove et al., 2005). To model additive compensation without overwriting, we retain an identity bypass and append a visually guided compensator:

$$Y_{\text{comp}} = \big(A_{\text{sel}}, \, \mathcal{C}(A_{\text{sel}}, V)\big). \qquad (3)$$

By the chain rule,

$$\begin{aligned} I(S; Y_{\text{comp}}) &= I\big(S; A_{\text{sel}}\big) + I\big(S; \mathcal{C}(A_{\text{sel}}, V) \mid A_{\text{sel}}\big) \\ &\geq I\big(S; A_{\text{sel}}\big), \end{aligned} \qquad (4)$$

where the inequality is strict when the conditional term is positive. We use this relation as a design motivation rather than a performance guarantee: the bypass preserves access to $A_{\text{sel}}$, while $\mathcal{C}(\cdot)$ can contribute complementary cues when helpful. From a predictive coding view, we encourage conservative refinement with residual $e = A_{\text{sel}} - \hat{A}_{\text{sel}}$,

$$\mathcal{E} = \mathcal{L}_{\text{rec}}(S, \hat{S}) + \lambda \|e\|^2, \qquad (5)$$

so the penalty favors small, error-reducing corrections.

For aligned visual features, we prefer augmentation over replacement. With $\tilde{V} = (V, f(V, Z_{\text{sel}}))$, the chain rule gives

$$\begin{aligned} I(S; \tilde{V}) &= I(S; V) + I\big(S; f(V, Z_{\text{sel}}) \mid V\big) \\ &\geq I(S; V), \end{aligned} \qquad (6)$$

motivating the use of aligned articulatory cues to supplement, not discard, the original visual stream.

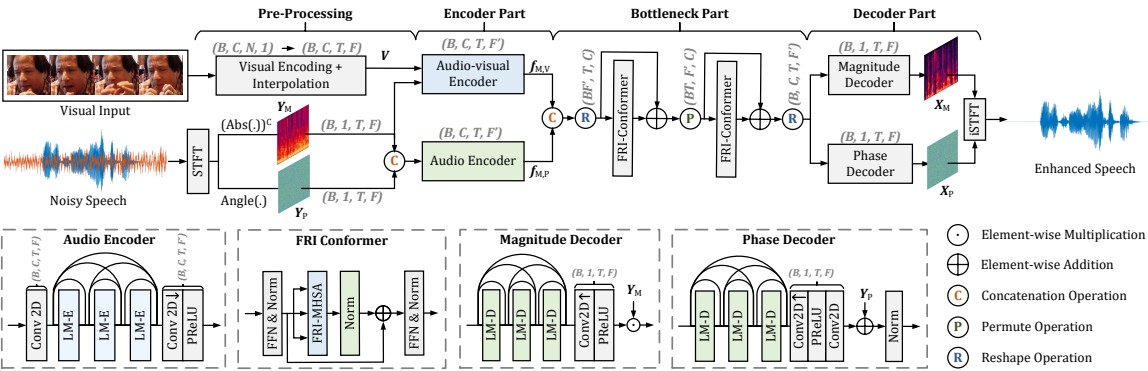

*Figure 2.* Overall framework of Neuro-SCNet. The architecture adopts a dual-encoder–decoder design: the audio encoder preserves acoustic fidelity, the audio–visual encoder integrates multimodal streams and incorporates auditory selection and cross-modal compensation, the FRI-Conformer bottleneck refines features via information recycling, and the dual decoders reconstruct clean speech.

In practice, attentional gating is implemented as an information-bottleneck gate with matched forward–backward suppression, while compensatory fusion uses an identity bypass with a residual objective that remains non-degrading under small updates. Aligned visual features are added by augmentation so that new cues increase, rather than replace, existing information.

### 3.2. Architecture Overview

As shown in Figure 2, Neuro-SCNet is a dual-encoder–decoder for estimating clean speech $\mathbf{x} \in \mathbb{R}^L$ from a noisy mixture $\mathbf{y} \in \mathbb{R}^L$ with visual guidance. The pipeline has four stages: pre-processing, encoding, a lightweight bottleneck with selection–compensation, and magnitude/phase decoding.

**Pre-processing.** We compute STFT on $\mathbf{y}$ to obtain magnitude $\mathbf{Y}'_m \in \mathbb{R}^{B \times 1 \times T \times F}$ and phase $\mathbf{Y}_p$, then apply power-law compression $\mathbf{Y}_m = (\mathbf{Y}'_m)^c$. Video frames are embedded by CTCNet-Lip (Li et al., 2024a) into $\mathbf{v}_{in} \in \mathbb{R}^{B \times N_v \times T_v}$. A $1 \times 1$ temporal projection and interpolation bring them to the audio grid ($T$ frames):

$$\hat{\mathbf{v}} = \text{Interp}\big(\text{Conv1D}(\mathbf{v}_{in})\big) \in \mathbb{R}^{B \times D \times T}. \quad (7)$$

As an *auxiliary* step, we estimate a small global offset in a bounded window and expose a confidence for later gating. A shallow audio track is formed by frequency averaging and temporal projection,

$$\mathbf{a} = \text{Conv1D}\big(\text{Avg}_F(\mathbf{Y}_m)\big) \in \mathbb{R}^{B \times D \times T}, \quad (8)$$

and both $\mathbf{a}$ and $\hat{\mathbf{v}}$ are $\ell_2$-normalized over $D$. For integer lag $\delta \in [-K, K]$,

$$c(\delta) = \frac{1}{T - |\delta|} \sum_{t=1}^{T-|\delta|} \langle \mathbf{a}_{:,t}, \hat{\mathbf{v}}_{:,t+\delta} \rangle, \quad (9)$$

$$p(\delta) = \frac{e^{\kappa c(\delta)}}{\sum_{\delta'=-K}^{K} e^{\kappa c(\delta')}}, \quad (10)$$

with soft offset $\hat{\delta} = \sum_\delta \delta \, p(\delta)$ and differentiable shifter $\tilde{\mathbf{v}} = \mathcal{S}_{\hat{\delta}}(\hat{\mathbf{v}})$. We define a reliability scalar $r = \max_\delta p(\delta) \in (0, 1]$ to down-weight visual conditioning when synchrony is weak. Subsequent fusion receives $(\mathbf{Y}_m, \mathbf{Y}_p, \tilde{\mathbf{v}}, r)$ and applies reliability gating on the visual pathway, e.g.

$$\mathbf{z}_t = \Phi\big(\mathbf{a}_{:,t}, \, r \cdot \tilde{\mathbf{v}}_{:,t}\big), \quad (11)$$

where $r$ is a clip-level synchrony confidence that coarsely modulates visual conditioning, rather than a dense frame-level or spatial reliability map. Implementation details of visual pre-alignment are provided in **Appendix Sec. A**.

**Encoders.** Two parallel encoders operate at the matched frame rate $T$. (i) *Audio encoder:* a compact convolutional stack processes $(\mathbf{Y}_m, \mathbf{Y}_p)$ to preserve fine-scale acoustics (harmonics, formants, temporal fine structure). (ii) *Audio–visual encoder:* a semantic branch extracts multimodal features and hosts the selection (ASM) and compensation (CCM) interfaces used in the bottleneck.

**Bottleneck.** A small stack of FRI-Conformer blocks (Xu et al., 2025b) refines the two streams. Within these blocks, ASM applies pre-fusion gating on the audio features to suppress non-target regions; CCM then injects visual articulatory cues under an identity bypass with residual correction. The bottleneck is lightweight by design and does not alter the auxiliary pre-alignment; the reliability scalar $r$ simply moderates visual influence.

**Magnitude and phase decoders.** Two decoders run in parallel. The magnitude branch estimates a mask $\mathbf{M}_m$ applied to $\mathbf{Y}_m$; the phase branch estimates a residual $\Delta \mathbf{Y}_p$ added to $\mathbf{Y}_p$. The enhanced spectrogram and refined phase are inverted by iSTFT to produce $\hat{\mathbf{x}}$. This dual path preserves low-level fidelity while benefiting from semantic selection–compensation.

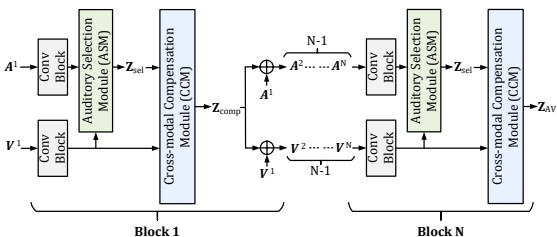

*Figure 3.* Architecture of the Audio–Visual Encoder. A stack of $N$ blocks, each with an Auditory Selection Module (ASM) and a Cross-Modal Compensation Module (CCM).

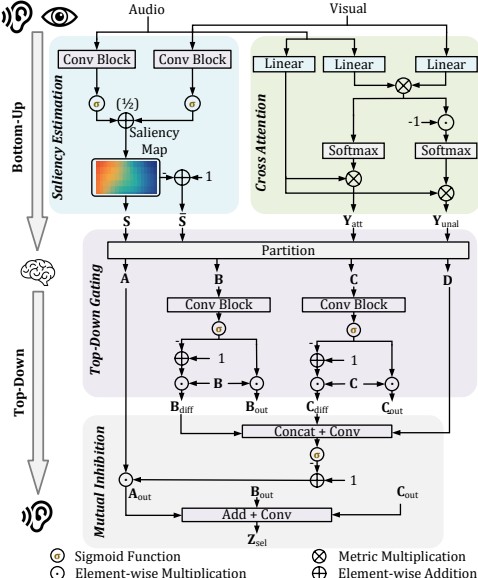

*Figure 4.* Architecture of the Auditory Selection Module (ASM). Audio and visual features form saliency maps and a joint mask. Cross-attention splits the audio into aligned and anti-aligned streams, partitioned into four parts (A–D); gates refine B and C, and a residual path inhibits distractors in A.

### 3.3. Audio-visual Encoder

As shown in Figure 3, the audio-visual encoder is realized as a stack of $N$ sequential blocks. Each block integrates modality-specific convolutional feature extraction with two core mechanisms: *Auditory Selection* and *Cross-Modal Compensation*. Through iterative refinement, the encoder enforces semantic alignment across modalities and recovers weak auditory cues, culminating in the final fused representation $f_{AV}$ from the last block.

**Auditory Selection Module.** The goal of auditory selection is to enhance target-consistent auditory evidence while suppressing irrelevant components, combining bottom-up saliency detection with top-down gating control. As shown in Figure 4, let $\mathbf{A}^{(i)}, \mathbf{V}^{(i)} \in \mathbb{R}^{B \times C \times F \times T}$ denote audio and visual features entering block $i$. A bottom-up stage first

estimates modality-specific saliency maps,

$$\mathbf{S}_a = \sigma(f(\mathbf{A}^{(i)})), \quad \mathbf{S}_v = \sigma(f(\mathbf{V}^{(i)})), \quad (12)$$

where $f(\cdot)$ is a two-layer $1 \times 1$ convolution with ReLU. These are fused into a joint saliency mask,

$$\mathbf{S} = \tfrac{1}{2}(\mathbf{S}_a + \mathbf{S}_v), \quad \bar{\mathbf{S}} = \mathbf{1} - \mathbf{S}, \quad (13)$$

which captures stimulus-driven prominence from both modalities. A cross-attention operator further provides bottom-up semantic alignment between modalities:

$$\mathbf{Y}_{att} = \mathrm{MHA}(Q = \mathbf{V}^{(i)}, K = \mathbf{A}^{(i)}, V = \mathbf{A}^{(i)}), \quad (14)$$

while an anti-aligned counterpart is computed as

$$\mathbf{P}_{unal} = \mathrm{softmax}\left(-\frac{QK^\top}{\sqrt{C}}\right), \quad \mathbf{Y}_{unal} = \mathbf{P}_{unal}\mathbf{A}^{(i)}. \quad (15)$$

The anti-aligned stream is used as a relative inconsistency cue for suppressive gating, rather than as an explicit semantic estimate of distractors. The top-down stage uses visual priors to regulate auditory feature selection. Partitioning with $\mathbf{S}$ yields four quadrants:

$$\mathbf{A} = \mathbf{S} \odot \mathbf{Y}_{att}, \qquad \mathbf{B} = \bar{\mathbf{S}} \odot \mathbf{Y}_{att}, \quad (16)$$

$$\mathbf{C} = \mathbf{S} \odot \mathbf{Y}_{unal}, \qquad \mathbf{D} = \bar{\mathbf{S}} \odot \mathbf{Y}_{unal}, \quad (17)$$

where $\mathbf{A}$ encodes salient aligned evidence, $\mathbf{B}$ non-salient aligned, $\mathbf{C}$ salient anti-aligned, and $\mathbf{D}$ non-salient anti-aligned. Learnable gates refine $\mathbf{B}$ and $\mathbf{C}$,

$$\mathbf{B}_{out} = \mathbf{M}_B \odot \mathbf{B}, \quad \mathbf{B}_{diff} = (\mathbf{1} - \mathbf{M}_B) \odot \mathbf{B}, \quad (18)$$

$$\mathbf{C}_{out} = \mathbf{M}_C \odot \mathbf{C}, \quad \mathbf{C}_{diff} = (\mathbf{1} - \mathbf{M}_C) \odot \mathbf{C}, \quad (19)$$

with gating masks $\mathbf{M}_B = \sigma(g_B(\mathbf{B})), \mathbf{M}_C = \sigma(g_C(\mathbf{C}))$ implemented by $1 \times 1$ convolutions with normalization and PReLU. Residual components from $\mathbf{B}_{diff}, \mathbf{C}_{diff}, \mathbf{D}$ inhibit $\mathbf{A}$ via a top-down control gate:

$$\mathbf{A}_{out} = (\mathbf{1} - \sigma(g_A([\mathbf{B}_{diff}, \mathbf{C}_{diff}, \mathbf{D}]))) \odot \mathbf{A}. \quad (20)$$

The final output is obtained as

$$\mathbf{Z}_{sel} = \mathrm{Conv}_{1 \times 1}(\mathbf{A}_{out} + \mathbf{B}_{out} + \mathbf{C}_{out}). \quad (21)$$

This formulation integrates bottom-up saliency detection and semantic alignment with top-down gating and inhibition. From an information-theoretic perspective (Sec. 3.1), it realizes the information bottleneck by maximizing $I(\mathbf{Z}_{sel}; S)$ while constraining redundancy. Neuro-cognitively, it corresponds to selective attention in the superior temporal gyrus (STG), where stimulus-driven responses are dynamically modulated by visual priors to suppress irrelevant auditory streams through mutual inhibition. (Fritz et al., 2007; Reynolds & Heeger, 2009).

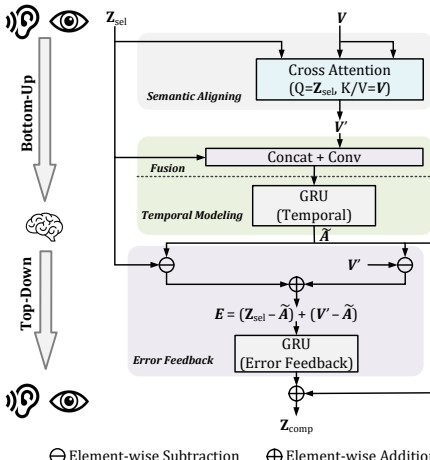

*Figure 5.* Architecture of the Cross-Modal Compensation Module (CCM). Visual cues are aligned to the selected audio with cross attention, fused, and then refined by a GRU with error-driven feedback. The output is a compensated audio representation.

**Cross-Modal Compensation.** While auditory selection improves discriminability, it may also suppress weak but useful cues. Cross-modal compensation restores such information by exploiting visual guidance, beginning with a cross-attention step that aligns video features to the auditory stream. Specifically, as shown in Figure 5, given the selected auditory representation $\mathbf{Z}_{\text{sel}}$ and the visual input $\mathbf{V}^{(i)}$, we compute

$$\mathbf{V}' = \text{MHA}(Q = \mathbf{Z}_{\text{sel}}, \, K = \mathbf{V}^{(i)}, \, V = \mathbf{V}^{(i)}), \quad (22)$$

where multi-head attention projects $\mathbf{V}^{(i)}$ into an aligned representation $\mathbf{V}'$ under the guidance of $\mathbf{Z}_{\text{sel}}$. This ensures that subsequent compensation operates on semantically synchronized embeddings. We then fuse the selected auditory feature and the reliability-gated visual feature,

$$\mathbf{U} = \phi([\mathbf{Z}_{\text{sel}}, r \cdot \mathbf{V}']), \quad (23)$$

where $\phi$ is a $1 \times 1$ convolution with nonlinearity. A temporal GRU produces an initial compensation estimate $\tilde{\mathbf{A}}$, and prediction errors are defined as

$$\mathbf{E} = (\mathbf{Z}_{\text{sel}} - \tilde{\mathbf{A}}) + r \cdot (\mathbf{V}' - \tilde{\mathbf{A}}). \quad (24)$$

A second GRU processes $\mathbf{E}$ to generate feedback $\mathbf{F}$, from which we obtain a residual compensation term,

$$\Delta \mathbf{Z} = \text{LN}(\mathbf{F}). \quad (25)$$

The final compensated representation is formed through an explicit identity bypass:

$$\mathbf{Z}_{\text{comp}} = \mathbf{Z}_{\text{sel}} + r \cdot \Delta \mathbf{Z}. \quad (26)$$

The representation is then passed to the subsequent block. Following Sec. 3.1, we augment the visual stream with an alignment-derived component, $\tilde{\mathbf{V}} = (\mathbf{V}, f(\mathbf{V}, \mathbf{Z}_{\text{sel}}))$, which satisfies Eq. 6. A deterministic projection $\mathbf{V}'$ does not, in general, increase mutual information, so it is used only as an intermediate to construct the augmentation rather than as evidence of information gain. From a neurocognitive standpoint, this procedure is consistent with findings on the superior temporal sulcus, where audio and visual signals are first temporally aligned and visual articulatory cues then support the recovery of missing auditory content (Beauchamp, 2005).

By stacking such blocks, the encoder progressively aligns semantics and enhances robustness. The final block's $\mathbf{Z}_{\text{comp}}$ is taken as the fused audio-visual embedding $f_{\text{AV}}$ for bottleneck processing.

## 4. Experiments

### 4.1. Datasets

We conduct experiments on widely used AVSS benchmarks constructed from three publicly available datasets: LRS2 (Afouras et al., 2018a), LRS3 (Afouras et al., 2018d), and VoxCeleb2 (Chung et al., 2018). We construct mixtures *within* the official corpus partitions released by each dataset to prevent speaker leakage and to ensure reproducibility across models. We follow commonly used AVSS mixture protocols on these corpora (Li et al., 2024b;a; Pegg et al., 2024). All audio clips are sampled at 16 kHz. For each mixture, two utterances from different speakers are randomly selected and combined at a signal-to-interference ratio (SIR) sampled uniformly from $[-5, 5]$ dB. Visual streams are synchronized with audio at 25 FPS, and mouth region frames are resized to $88 \times 88$ pixels grayscale images. Details on the datasets and on the three- and four-speaker mixture settings are given in **Appendix Sec. B**.

### 4.2. Training Details and Evaluation Metrics

All models were implemented in PyTorch and trained on six NVIDIA RTX 3090 GPUs (24 GB each). Networks were optimized from scratch using Adam (Ai & Ling, 2023) with an initial learning rate of $5 \times 10^{-5}$, $\beta_1 = 0.9$, $\beta_2 = 0.999$, and weight decay of $1 \times 10^{-4}$. The learning rate was reduced linearly once the validation performance plateaued. Training used a batch size of 64, with each epoch comprising 64K randomly sampled pairs. Training lasted 250 epochs. To speed up convergence, the visual front-end was frozen, and its features were pre-extracted and cached. For offset estimation we use $K = 5$ (about $\pm 200$ ms at 25 FPS), with temperature parameter $\kappa$ selected on a validation set (initialized in $[6, 10]$ and optionally annealed). The cross-correlation is computed once before fusion with complexity $O(TCK)$. If fractional shifts are not desired, we use the integer maximizer $\arg\max_\delta p(\delta)$ for $\mathcal{S}_{\hat{\delta}}$ while still computing

*Table 1.* Performance of different AVSS models on LRS2, LRS3, and VoxCeleb2. Results are averaged across all test speakers, with larger SI-SNRi, SDRi, and PESQ values indicating better quality. A dash "–" indicates that the metric is not reported in the original paper.

| Model | Params(M) | LRS2 | | | LRS3 | | | VoxCeleb2 | | |
|---|---|---|---|---|---|---|---|---|---|---|
| | | SI-SNRi (dB) | SDRi (dB) | PESQ | SI-SNRi (dB) | SDRi (dB) | PESQ | SI-SNRi (dB) | SDRi (dB) | PESQ |
| AV-ConvTasNet (Wu et al., 2019) | 16.45 | 12.5 | 12.8 | 2.69 | 11.2 | 11.7 | 2.58 | 9.2 | 9.8 | 2.17 |
| VisualVoice (Gao & Grauman, 2021) | 77.8 | 11.5 | 11.8 | 2.78 | 9.9 | 10.3 | 2.13 | 9.3 | 10.2 | 2.45 |
| CaffNet-C (Lee et al., 2021) | - | - | 10.0 | 1.15 | - | 9.8 | - | - | 7.6 | - |
| CTC-Net (Li et al., 2024a) | 7.0 | 14.3 | 14.6 | 3.08 | 17.4 | 17.5 | 3.24 | 11.9 | 13.1 | 3.00 |
| AVLiT-8 (Martel et al., 2023) | 5.75 | 12.8 | 13.1 | 2.56 | 13.5 | 13.6 | 2.78 | 9.4 | 9.9 | 2.23 |
| RTFS-Net-12 (Pegg et al., 2024) | **0.7** | 14.9 | 15.1 | 3.07 | 17.5 | 17.6 | 3.25 | 12.4 | 13.6 | 3.00 |
| IIANet (Li et al., 2024b) | 3.1 | 16.0 | 16.2 | 3.23 | 18.3 | 18.5 | 3.28 | 13.6 | 14.3 | 3.12 |
| AV-CrossNet (Kalkhorani et al., 2025) | 11.1 | 16.8 | 17.1 | 3.56 | 18.3 | 18.5 | 3.67 | 14.6 | 14.9 | 3.41 |
| Neuro-SCNet (Proposed) | 6.3 | **17.2** | **17.9** | **3.59** | **18.9** | **19.5** | **3.71** | **14.8** | **15.2** | **3.43** |

*Table 2.* Performance of AVSS models on LRS2 mixtures with two to four overlapping speakers, reported in terms of SI-SNRi.

| Methods | LRS2-2Mix | LRS2-3Mix | LRS2-4Mix |
|---|---|---|---|
| AV-ConvTasNet (Wu et al., 2019) | 12.5 | 7.8 | 3.6 |
| AVLiT-8 (Martel et al., 2023) | 12.8 | 9.9 | 5.2 |
| CTCNet (Li et al., 2024a) | 14.3 | 11.2 | 6.6 |
| IIANet (Li et al., 2024b) | 16.0 | 12.8 | 8.2 |
| Neuro-SCNet (Proposed) | **17.2** | **13.9** | **9.8** |

*Table 3.* Model sizes, computational complexity, and inference efficiency of various AVSS methods, evaluated on 1-second 16 kHz audio and 25 FPS video inputs. Inference was performed on an NVIDIA RTX 3090 GPU and an Intel Xeon Platinum 8269CY CPU. A dash "–" indicates that the metric is not reported in the original paper and no public code or checkpoints are available to reproduce it under our setup.

| Model | Params.(M) | MACs(G) | Inference Time | |
|---|---|---|---|---|
| | | | GPU(ms) | CPU(s) |
| AV-ConvTasNet (Wu et al., 2019) | 16.5 | 23.8 | 118.77 | 1.22 |
| VisualVoice (Gao & Grauman, 2021) | 77.8 | **9.7** | 231.65 | 3.04 |
| CTC-Net (Li et al., 2024a) | 7.0 | 167.1 | 162.45 | 1.69 |
| AVLiT-8 (Martel et al., 2023) | 5.8 | 18.2 | **116.27** | **1.15** |
| RTFS-Net-12 (Pegg et al., 2024) | **0.7** | 56.4 | 144.61 | 1.52 |
| IIANet (Li et al., 2024b) | 3.1 | 18.6 | 238.94 | 1.46 |
| AV-CrossNet (Kalkhorani et al., 2025) | 11.1 | - | - | - |
| Neuro-SCNet | 6.3 | 17.8 | 128.87 | 1.39 |

$r$ from the soft distribution.

We adopt a multi-level loss function comprising SI-SNR loss (Hu et al., 2023) ($\mathcal{L}_{sisnr}$), magnitude loss ($\mathcal{L}_M$), phase loss ($\mathcal{L}_P$) (Lu et al., 2025; Ai & Ling, 2023), and complex loss ($\mathcal{L}_C$). The total loss is computed as:

$$\mathcal{L}_{Total} = 0.025\mathcal{L}_{sisnr} + 0.9\mathcal{L}_M + 0.1\mathcal{L}_C + 0.3\mathcal{L}_P. \quad (27)$$

The weights for $\mathcal{L}_M$, $\mathcal{L}_C$, and $\mathcal{L}_P$ follow (Lu et al., 2025), while the weight for $\mathcal{L}_{sisnr}$ is scaled to match the range of the other terms.

For evaluation, we report SI-SNR improvement (SI-SNRi), signal-to-distortion ratio improvement (SDRi) (Le Roux et al., 2019), and perceptual evaluation of speech quality (PESQ) (Rix et al., 2001) on the test set. SI-SNRi and SDRi are computed as improvements over the input mixture using the same reference alignment and normalization policy across all methods. PESQ is computed in wideband mode at 16 kHz on the aligned estimates.

### 4.3. Model Comparison

Table 1 shows results for Neuro-SCNet and the baselines AV-ConvTasNet (Wu et al., 2019), VisualVoice (Gao & Grauman, 2021), CaffNet-C (Lee et al., 2021), CTC-Net (Li et al., 2024a), AVLiT-8 (Martel et al., 2023), RTFS-Net (Pegg et al., 2024), IIANet (Li et al., 2024b), and AV-CrossNet (Kalkhorani et al., 2025). Neuro-SCNet achieves the best SI-SNRi, SDRi, and PESQ on all three datasets. Compared with AV-CrossNet, the strongest prior model, Neuro-SCNet improves LRS3 by 0.6 dB in SI-SNRi and by 0.04 in PESQ, and matches or exceeds it on LRS2 and VoxCeleb2. Relative to the neuroscience-inspired CTC-Net and IIANet, Neuro-SCNet shows consistent gains in both SI-SNRi and PESQ. The model uses 6.3M parameters, giving a strong accuracy–efficiency balance.

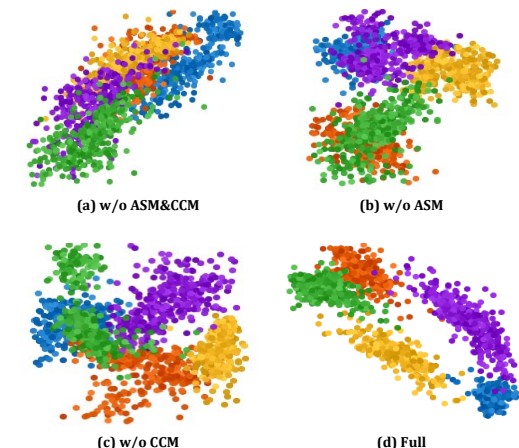

*Figure 6.* t-SNE visualization of speaker embeddings from separated speech under different ablation settings. Each point denotes an utterance embedding from one of five speakers, with colors indicating speaker identity.

To test robustness under stronger interference, we extend LRS2 to three- and four-speaker mixtures, denoted LRS2-3Mix and LRS2-4Mix, following the same mixing protocol and SNR range as prior work (Li et al., 2024b). Table 2 shows that Neuro-SCNet ranks first at both overlap levels. On the four-speaker condition, it exceeds IIANet by 1.6 dB in SI-SNRi. Table 3 summarizes complexity: 6.3M parameters and 17.8G MACs with fast CPU inference. Models with

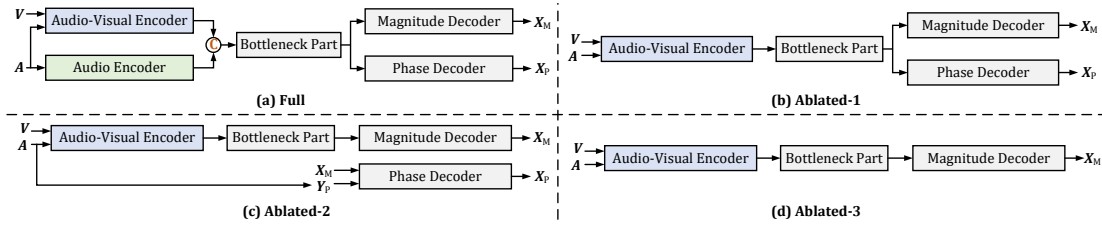

*Figure 7.* Overview of the full Neuro-SCNet and its encoder–decoder ablations. (a) Full: dual encoders with parallel magnitude and phase decoders. (b) Ablated-1: audio encoder removed, joint audio–visual encoder only. (c) Ablated-2: phase subnetwork replaces the phase-aware path. (d) Ablated-3: no magnitude–phase joint modeling, single magnitude decoder.

*Table 4.* Module-level ablations of *Auditory Selection* (ASM) and *Cross-Modal Compensation* (CCM) on LRS2-2Mix/3Mix/4Mix in terms of SI-SNRi (dB).

| Variant | Params (M) | SI-SNRi (dB) | | |
|---|---|---|---|---|
| | | LRS2-2Mix | LRS2-3Mix | LRS2-4Mix |
| **Neuro-SCNet (Full)** | 6.3 | **17.2** | **13.9** | **9.8** |
| w/o ASM | 6.0 | 16.4 | 12.7 | 8.6 |
| w/o CCM | 6.0 | 16.2 | 12.5 | 8.4 |
| w/o ASM & CCM | 5.8 | 15.1 | 11.2 | 7.0 |

*Table 5.* Ablation study on the **encoder–decoder structure** of Neuro-SCNet, evaluated on the *LRS2* dataset in terms of SI-SNRi (dB). Each variant corresponds to the structural modifications illustrated in Figure 7.

| Method / Variant | LRS2-2Mix | LRS2-3Mix | LRS2-4Mix |
|---|---|---|---|
| **Neuro-SCNet (Full)** | **17.2** | **13.9** | **9.8** |
| *Phase assumptions* | | | |
| Replaced with mixture phase | 14.8 | 11.3 | 6.5 |
| Replaced with oracle clean phase | 17.5 | 14.2 | 10.0 |
| *Structural ablations (cf. Figure 7)* | | | |
| Ablated-1 (remove audio encoder) | 16.0 | 12.1 | 8.1 |
| Ablated-2 (phase sub-network) | 16.2 | 12.6 | 8.6 |
| Ablated-3 (time-domain modeling) | 16.3 | 11.8 | 8.7 |
| Ablated-3 (complex-domain modeling) | 15.4 | 11.5 | 8.5 |

*Table 6.* Time-shift robustness (SI – SNRi, dB) with $\Delta$ (ms) on the horizontal axis.

| $\Delta$ (ms) | $-200$ | $-100$ | 0 | 100 | 200 |
|---|---|---|---|---|---|
| No-offset | 14.8 | 15.8 | 16.6 | 15.2 | 14.8 |
| Offset-only | 14.6 | 16.3 | 16.9 | 16.2 | 14.6 |
| Offset+Reliability (Default) | 16.2 | 16.9 | 17.2 | 16.8 | 16.2 |

much lower compute are faster but lose accuracy, which is consistent with the trends in Tables 1 and 2.

### 4.4. Ablation Study

We evaluate the contribution of each component on LRS2 using two-, three-, and four-speaker mixtures (2Mix, 3Mix, 4Mix) and report SI-SNRi, SDRi, and PESQ.

**Ablations of ASM and CCM.** Table 4 reports the impact of removing the auditory selection module (ASM) and the cross-modal compensation module (CCM). The full Neuro-SCNet consistently outperforms all ablated variants on LRS2-2Mix/3Mix/4Mix. Eliminating ASM reduces the ability to suppress distractors under visual guidance, while removing CCM limits residual refinement through visual cues. The performance gap becomes larger as the number of speakers increases, indicating that both modules are indispensable and complementary rather than redundant. Figure 6 further visualizes speaker embeddings of separated speech. Without ASM and CCM, clusters overlap heavily. Removing only ASM yields partially separated but poorly defined clusters. Removing only CCM produces clusters that are discernible yet internally dispersed. In contrast, the full model produces well-separated and compact clusters.

**Encoder–Decoder Ablation.** Figure 7 and Table 5 evaluate encoder–decoder choices on LRS2-2Mix, 3Mix, and 4Mix. The full Neuro-SCNet with a dual encoder and parallel magnitude–phase decoders gives the highest SI-SNRi in all cases. Removing the audio encoder (Ablated-1) lowers accuracy because the model loses fine acoustic cues needed for stable reconstruction. Using a separate phase subnetwork (Ablated-2) recovers part of the loss but remains below the

full model, showing that joint magnitude–phase modeling is more effective than treating them independently. Switching to a single-domain design (Ablated-3) causes the largest drop, especially at higher overlap. These results indicate that the dual encoder is key for keeping semantic alignment and acoustic detail, and that parallel magnitude–phase decoding is important for robust separation under heavy interference.

**Time-Shift Robustness of Visual Pre-Alignment.** We inject a visual shift $\Delta \in [-200, 200]$ ms at test time while keeping model weights fixed, changing only the visual pre-processing: *No-offset* uses interpolation only, *Offset-only* estimates and applies a shift, and *Offset+Reliability* further weights the visual path by confidence (default). Table 6 shows a clear trend: No-offset peaks at zero shift and declines as $|\Delta|$ increases; Offset-only improves scores near zero shift but converges to No-offset at large shifts; Offset+Reliability is highest for all $\Delta$, preserving gains when alignment is accurate and reducing the impact of unreliable visuals under large misalignment.

## 5. Conclusion

We present Neuro-SCNet, an AVSS model with two explicit stages. The auditory selection stage applies early gating

to audio features before fusion. The cross-modal compensation stage integrates aligned visual cues with residual feedback to recover missing structure. A dual encoder preserves high-level cues and low-level acoustics, and parallel magnitude–phase decoders support accurate reconstruction. On LRS2, LRS3, and VoxCeleb2, the model achieves state-of-the-art SI-SNRi, SDRi, and PESQ. Performance remains first under three- and four-speaker overlap, and reliability-gated visual pre-alignment yields stable results under controlled time shifts. Ablations verify that both selection and compensation are necessary and that magnitude–phase decoding further improves quality.

## Impact Statement

This work aims to improve audio-visual speech separation in noisy multi-speaker environments, with potential benefits for hearing assistance, online communication, and robust speech interfaces. Since the method uses face-conditioned visual speech cues, deployment should be limited to settings with appropriate consent, privacy protection, and safeguards against non-consensual speech extraction or surveillance-oriented use. The model may also be less reliable under severe mouth-region occlusion or inaccurate face tracking, and future work should develop finer-grained visual reliability estimation for safer deployment.

## Acknowledgements

This work has been fully supported by the Research Impact Fund by the Research Grants Council of Hong Kong (Project No. 130272) and a grant from the Research Grants Council of the Hong Kong Special Administrative Region, China (R1015-23); the Faculty Research Grants (SDS24A8, SDS25A15 and SDS24A19), the Interdisciplinary & Strategic Research Grant (ISRG252606), and the Direct Grants (DR25E8 and DR26F2) of Lingnan University, Hong Kong.

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

## A. Implementation of Visual Pre-alignment

This section specifies the visual pre-alignment exactly as implemented in Algorithm 1. The audio magnitude spectrogram is $\mathbf{Y}_m \in \mathbb{R}^{B \times 1 \times T \times F}$ with $F{=}257$, and the video embeddings are $\mathbf{v}_{\text{in}} \in \mathbb{R}^{B \times C_v \times T_v}$. The video stream is first resized to the audio time grid by bilinear interpolation with `align_corners` = False and temporally given a singleton width, then mapped to the frequency axis by a $3{\times}1$ Conv2D:

$$\mathbf{v}_{\text{map}} \in \mathbb{R}^{B \times F \times T} \leftarrow \text{Conv2D}_{3 \times 1}\big(\text{Interp}_{\text{bilinear}}(\mathbf{v}_{\text{in}})\big). \tag{28}$$

The audio track for correlation uses the magnitude spectrogram rearranged as $\mathbf{a}_{\text{spec}} = \text{permute}(\mathbf{Y}_m) \in \mathbb{R}^{B \times F \times T}$, where the singleton channel is removed and $(T, F)$ are swapped to $(F, T)$.

Both streams are projected to a $D$-channel semantic space by $1{\times}1$ Conv1D along the frequency axis and then $\ell_2$-normalized over channels, which implements a learnable frequency aggregation rather than a fixed average:

$$\mathbf{a}_{\text{sem}} = \text{Norm}_{\ell_2}\big(\text{Conv1D}_{F \to D}(\mathbf{a}_{\text{spec}})\big), \tag{29}$$

$$\mathbf{v}_{\text{sem}} = \text{Norm}_{\ell_2}\big(\text{Conv1D}_{F \to D}(\mathbf{v}_{\text{map}})\big), \tag{30}$$

with $\mathbf{a}_{\text{sem}}, \mathbf{v}_{\text{sem}} \in \mathbb{R}^{B \times D \times T}$ and $D{=}64$ in the default setting.

A bounded search over integer lags $\delta \in [-K, K]$ is performed by time-overlapped dot products of the normalized semantic trajectories, averaged over channels and time to produce a correlation curve $c(\delta) \in \mathbb{R}^{B \times (2K+1)}$:

$$c(\delta) = \frac{1}{T - |\delta|} \sum_{t=1}^{T-|\delta|} \frac{1}{D} \big\langle \mathbf{a}_{\text{sem}}[:, :, t], \ \mathbf{v}_{\text{sem}}[:, :, t + \delta] \big\rangle. \tag{31}$$

A soft distribution over lags is obtained by a temperature parameter $\kappa > 0$,

$$p(\delta) = \frac{\exp\{\kappa\, c(\delta)\}}{\sum_{\delta'=-K}^{K} \exp\{\kappa\, c(\delta')\}}, \tag{32}$$

and the soft offset is the expectation

$$\hat{\delta} = \sum_{\delta=-K}^{K} \delta\, p(\delta). \tag{33}$$

The visual map $\mathbf{v}_{\text{map}}$ is shifted along time by a differentiable operator implemented with `grid_sample` in bilinear mode and `padding_mode="border"`. Time indices are normalized to $[-1, 1]$ consistently with the `align_corners` setting in the module. Denoting the shifted sequence by $\tilde{\mathbf{v}}_{\text{map}} \in \mathbb{R}^{B \times F \times T}$, the tensor is finally rearranged back to the audio layout

$$\tilde{\mathbf{v}} = \text{permute}(\tilde{\mathbf{v}}_{\text{map}}) \in \mathbb{R}^{B \times 1 \times T \times F}, \tag{34}$$

which exactly matches the shape of $\mathbf{Y}_m$ for downstream fusion.

Reliability is a scalar derived from the peak probability of the lag distribution and is used later as a broadcast gate. The implementation takes

$$r = \max_{\delta} p(\delta) \in (0, 1], \tag{35}$$

and returns its broadcast form $r_{\text{bc}} \in \mathbb{R}^{B \times 1 \times 1 \times 1}$.

The module outputs only the shifted visual tensor and the reliability factor,

$$(\tilde{\mathbf{v}}, r_{\text{bc}}) = \text{VPA}(\mathbf{Y}_m, \mathbf{v}_{\text{in}}), \tag{36}$$

and does not perform any reliability-weighted blending with the unshifted track inside pre-processing. Any gating or blending that uses $r_{\text{bc}}$ is applied in later fusion blocks. Typical hyperparameters are $F{=}257$, $D{=}64$, $K{=}5$, and $\kappa{=}8.0$. The computational cost scales as $\mathcal{O}(B\, D\, T\, (2K{+}1))$ with negligible memory beyond the correlation curve.

---

**Algorithm 1** Implementation-consistent VPA

---

**Input:** $\mathbf{Y}_m \in \mathbb{R}^{B \times 1 \times T \times F}$, $\mathbf{v}_{\text{in}} \in \mathbb{R}^{B \times C_v \times T_v}$, window $K$, temperature $\kappa$
**Output:** $\tilde{\mathbf{v}} \in \mathbb{R}^{B \times 1 \times T \times F}$, $r_{\text{bc}} \in \mathbb{R}^{B \times 1 \times 1 \times 1}$

1   $\mathbf{v}_{\text{map}} \leftarrow \text{Conv2D}_{3 \times 1}\big(\text{Interp}_{\text{bilinear, align\_corners}=False}(\mathbf{v}_{\text{in}})\big)$ to $(B, F, T)$
    $\mathbf{a}_{\text{spec}} \leftarrow \text{permute}(\mathbf{Y}_m)$ to $(B, F, T)$
    $\mathbf{a}_{\text{sem}} \leftarrow \text{Norm}_{\ell_2}\big(\text{Conv1D}_{F \rightarrow D}(\mathbf{a}_{\text{spec}})\big);$    $\mathbf{v}_{\text{sem}} \leftarrow \text{Norm}_{\ell_2}\big(\text{Conv1D}_{F \rightarrow D}(\mathbf{v}_{\text{map}})\big)$
    **for** $\delta = -K$ **to** $K$ **do**
2     $\lfloor \ c(\delta) \leftarrow \text{mean}_t\big(\frac{1}{D}\langle \mathbf{a}_{\text{sem}}[:,:,t], \ \mathbf{v}_{\text{sem}}[:,:,t+\delta]\rangle\big)$
3   $p(\delta) \leftarrow \text{softmax}_\delta\big(\kappa \, c(\delta)\big);$    $\hat{\delta} \leftarrow \sum_\delta \delta \, p(\delta)$
    $\tilde{\mathbf{v}}_{\text{map}} \leftarrow \text{grid\_sample}\big(\mathbf{v}_{\text{map}}, \hat{\delta}\big)$ with bilinear mode, padding_mode="border"
    $\tilde{\mathbf{v}} \leftarrow \text{permute}(\tilde{\mathbf{v}}_{\text{map}})$ to $(B, 1, T, F)$
    $r \leftarrow \max_\delta p(\delta);$    $r_{\text{bc}} \leftarrow r.\text{view}(B, 1, 1, 1)$

---

## B. Details of Dataset

This section summarizes the datasets, selection criteria, preprocessing, and mixture protocols used in our experiments. We use the official corpus partitions released with each dataset. For LRS2 and LRS3, mixtures are constructed within the official train/validation/test splits. For VoxCeleb2, we follow the official Dev/Test partition and construct training and validation mixtures within the Dev partition under fixed mixture lists.

**LRS2 (Lip Reading Sentences 2)** (Afouras et al., 2018a). LRS2 comprises video clips sourced from BBC broadcasts and naturally includes background noise and room reverberation. Following commonly used AVSS protocols (Li et al., 2024b;a; Pegg et al., 2024), we construct a two-speaker mixture set (LRS2-2Mix) by randomly selecting two utterances from different speakers and mixing them at SIRs uniformly drawn from $[-5, 5]$ dB. The final split contains approximately 11 hours for training, 3 hours for validation, and 1.5 hours for testing.

**LRS3 (Lip Reading Sentences 3)** (Afouras et al., 2018d). LRS3 is derived from YouTube TED talks and generally provides cleaner audio than LRS2. We apply the same two-speaker mixture procedure as above. Owing to its lower ambient noise, LRS3 serves as a cleaner benchmark. The resulting split includes 28 hours of training, 3 hours of validation, and 1.5 hours of testing, consistent with recent AVSS evaluations (Li et al., 2024b;a; Pegg et al., 2024).

**VoxCeleb2** (Chung et al., 2018). VoxCeleb2 contains over one million utterances from more than 6,000 speakers recorded under diverse real-world conditions, with varying noise and reverberation. We randomly sample 5% of the official Dev partition and form two-speaker mixtures using the same protocol as for LRS2/LRS3, with SIRs uniformly drawn from $[-5, 5]$ dB. Training and validation mixtures are constructed within the sampled Dev subset under fixed mixture lists, and test mixtures are constructed on the official Test partition. The resulting configuration yields approximately 56 hours for training, 3 hours for validation, and 1.5 hours for testing.

**Multi-speaker configuration**. To assess scalability to more complex scenes, we further create three-speaker (LRS2-3Mix) and four-speaker (LRS2-4Mix) sets on LRS2, following (Li et al., 2024b). For each mixture, three or four distinct speakers are selected and their utterances are mixed at SIRs uniformly drawn from $[-5, 5]$ dB. Visual inputs are prepared using the same procedure as for LRS2-2Mix, with one target visual stream retained per mixture. Each of LRS2-3Mix and LRS2-4Mix contains 20,000 training samples, 5,000 validation samples, and 300 test samples. All baselines and Neuro-SCNet are trained on the same mixtures to ensure controlled comparisons across speaker counts.

## C. More Comparison Results

### C.1. Comparison with Low-quality Visual Input

To assess robustness under degraded visual conditions, we simulate three common corruptions on clean visual frames, following (Hong et al., 2023):

- **Occlusion:** randomly mask up to three axis-aligned rectangular regions per clip;

*Table 7.* Separation results on LRS2, LRS3, and VoxCeleb2 under three visual conditions: High-quality (clean), Occlusion, and Noise+Blur. All clips are 8 s; when occlusions are applied, 6 s (3 s at each side) are occluded.

| Model | LRS2 | | | LRS3 | | | VoxCeleb2 | | |
|---|---|---|---|---|---|---|---|---|---|
| | SI-SNRi (dB) | SDRi (dB) | PESQ | SI-SNRi (dB) | SDRi (dB) | PESQ | SI-SNRi (dB) | SDRi (dB) | PESQ |
| **High-quality visual input (clean)** | | | | | | | | | |
| AV-ConvTasNet-LQ (Wu et al., 2022) | 13.6 | 13.4 | 2.85 | 14.0 | 13.4 | 2.87 | 10.0 | 10.2 | 2.74 |
| MHSA-CRN (Xu et al., 2022) | 14.1 | 13.7 | 2.92 | 13.4 | 13.3 | 3.02 | 10.8 | 10.9 | 2.78 |
| RAVSS (Pan et al., 2024) | 14.2 | 14.5 | 3.06 | 14.6 | 14.8 | 3.08 | 12.4 | 12.5 | 3.02 |
| Neuro-SCNet (Proposed) | **17.2** | **17.9** | **3.59** | **18.9** | **19.5** | **3.71** | **14.8** | **15.2** | **3.43** |
| **Occlusion** | | | | | | | | | |
| AV-ConvTasNet-LQ (Wu et al., 2022) | 12.8 | 12.6 | 2.72 | 11.2 | 11.6 | 2.83 | 9.1 | 9.3 | 2.63 |
| MHSA-CRN (Xu et al., 2022) | 12.5 | 12.9 | 2.88 | 11.8 | 12.1 | 2.99 | 10.5 | 10.6 | 2.74 |
| RAVSS (Pan et al., 2024) | 13.1 | 13.9 | 3.00 | 14.3 | 14.5 | 3.03 | 11.7 | 11.8 | 2.93 |
| Neuro-SCNet (Proposed) | **15.0** | **15.6** | **3.48** | **16.3** | **16.9** | **3.59** | **12.9** | **13.2** | **3.35** |
| **Noise + Blur** | | | | | | | | | |
| AV-ConvTasNet-LQ (Wu et al., 2022) | 13.3 | 13.0 | 2.82 | 13.8 | 13.1 | 2.85 | 9.5 | 9.9 | 2.71 |
| MHSA-CRN (Xu et al., 2022) | 13.8 | 13.4 | 2.79 | 13.1 | 13.0 | 2.77 | 9.2 | 9.8 | 2.66 |
| RAVSS (Pan et al., 2024) | 13.8 | 14.1 | 3.01 | 14.0 | 14.3 | 3.04 | 12.0 | 12.2 | 2.99 |
| Neuro-SCNet (Proposed) | **15.7** | **16.4** | **3.51** | **16.7** | **17.3** | **3.62** | **13.5** | **13.7** | **3.37** |

- **Gaussian blur:** apply a $7 \times 7$ kernel with standard deviation $\sigma \sim \mathcal{U}(0.1, 2.0)$;

- **Gaussian noise:** add zero-mean noise with variance $\nu \sim \mathcal{U}(0, 0.2)$.

Parameters are sampled independently for each clip. We evaluate two regimes: (i) models trained on clean visual inputs (corruptions only at test time), and (ii) models trained with the same corruptions used at test time (corruption-matched training).

Table 7 summarizes performance on LRS2, LRS3, and VoxCeleb2 with clean visuals and with two forms of degradation, namely occlusion and the combination of Gaussian noise with blur. Across all datasets and metrics, Neuro-SCNet delivers the strongest separation quality with clean inputs and the advantage remains when the visual stream is corrupted. The gap is most visible on LRS3, where the audio and visual signals are well aligned and visual evidence is highly informative, while VoxCeleb2 exhibits lower absolute scores for every method due to greater variability, yet the ordering of systems is unchanged. The three baselines in the table reflect common choices for low-quality audio–visual separation. AV-ConvTasNet-LQ (Lee et al., 2021) adapts an audio-dominant architecture to degraded video and provides a conservative point of reference. MHSA-CRN (Jang et al., 2017) augments a convolutional backbone with multi-head channel attention and gains modestly on several metrics. RAVSS (Katsaggelos et al., 2015) adds residual aware visual selection and is typically the strongest among the baselines. Neuro-SCNet surpasses all three under clean conditions on each dataset and metric reported in the table.

Occlusion produces the largest drop for every model because mouth region evidence is partially removed rather than corrupted. Even under this stress, Neuro-SCNet maintains clear margins over AV-ConvTasNet-LQ, MHSA-CRN, and RAVSS on LRS2, LRS3, and VoxCeleb2. The persistence of the lead is consistent with the design choice of selection then compensation. Early gating limits unreliable visual influence when large regions are missing and prevents distractor driven updates in shared representations. The compensatory pathway contributes when useful visual context remains, which is less often the case under long occlusions. When degradation arises from Gaussian noise and blur, the impact is milder because articulatory cues are imprecise but still present. In this regime Neuro-SCNet preserves its lead on LRS2 and LRS3 and remains the top system on VoxCeleb2. Training the models with the same corruptions used at test time further improves results for all methods.

*Table 8.* Separation performance on NTCD-TIMIT and LRS3+WHAM! benchmarks, evaluated with PESQ, eSTOI, and SI-SDRi.

| Method | NTCD-TIMIT | | | LRS3+WHAM! | | |
|---|---|---|---|---|---|---|
| | PESQ | eSTOI | SI-SDRi | PESQ | eSTOI | SI-SDRi |
| Unprocessed | 1.19 | 0.33 | 0.0 | 1.08 | 0.37 | 0.0 |
| AV-ConvTasNet (Wu et al., 2019) | 1.33 | 0.40 | 9.02 | 1.29 | 0.60 | 6.21 |
| LAVSE (Chuang et al., 2020) | 1.31 | 0.37 | 6.22 | 1.24 | 0.50 | 5.59 |
| L2L (Ephrat et al., 2018) | 1.23 | 0.26 | 3.16 | 1.25 | 0.51 | 7.60 |
| VisualVoice (Gao & Grauman, 2021) | 1.45 | 0.43 | 10.04 | 1.48 | 0.62 | 11.84 |
| AVLiT-8 (Martel et al., 2023) | 1.43 | 0.45 | 11.00 | 1.52 | 0.65 | 12.01 |
| AV-CrossNet (Kalkhorani et al., 2025) | 2.16 | 0.62 | 15.60 | 2.03 | **0.77** | 13.84 |
| Neuro-SCNet (Proposed) | **2.18** | **0.65** | **15.94** | **2.08** | 0.77 | **13.90** |

*Table 9.* Target speaker extraction results on LRS3, TCD-TIMIT, and VoxCeleb2. All systems are trained with VoxCeleb2 using a unified visual frontend, and performance is reported in terms of SI-SDR (dB) and PESQ.

| Method | LRS3 | | TCD-TIMIT | | VoxCeleb2 | |
|---|---|---|---|---|---|---|
| | SI-SDR | PESQ | SI-SDR | PESQ | SI-SDR | PESQ |
| Unprocessed | 0.13 | 1.21 | -0.15 | 1.47 | -0.08 | 1.24 |
| VisualVoice (Gao & Grauman, 2021) | 11.60 | 2.27 | 10.88 | 2.25 | 9.73 | 1.97 |
| AV-ConvTasNet (Wu et al., 2019) | 12.13 | 2.33 | 11.53 | 2.21 | 10.38 | 1.97 |
| MuSE (Pan et al., 2021) | 12.97 | 2.56 | 12.50 | 2.25 | 11.24 | 2.20 |
| AV-SepFormer (Lin et al., 2023) | 13.81 | 2.67 | 13.44 | 2.57 | 12.13 | 2.31 |
| AVSepChain (Mu & Yang, 2024) | 15.20 | 3.12 | 14.70 | 2.88 | 12.40 | 2.72 |
| AV-CrossNet (Kalkhorani et al., 2025) | 17.42 | **3.14** | 18.15 | 3.25 | **14.71** | 2.93 |
| Neuro-SCNet (Proposed) | **18.04** | **3.14** | **18.31** | **3.26** | 14.66 | **2.95** |

## C.2. Comparison on AVSS with noise

To evaluate robustness when the visual stream is reliable but the acoustic scene is noisy, we conduct a noise–robustness study on NTCD-TIMIT and on a synthetic LRS3+WHAM! corpus, following (Kalkhorani et al., 2025). For NTCD-TIMIT (Abdelaziz, 2017), mixtures are formed by combining two utterances and adding background noise; the training, validation, and test partitions contain approximately five, one, and one hours of audio. The target–to–interferer ratio is fixed at 0 dB, and the added noise level is set by signal–to–noise ratios uniformly sampled from −5 to 20 dB. For LRS3+WHAM!, clean LRS3 speech is mixed with WHAM! noise (Wichern et al., 2019); mixtures use target–to–interferer ratios from −5 to 5 dB and signal–to–noise ratios from −6 to 3 dB, yielding about twenty–eight hours for training, three hours for validation, and two hours for testing.

Table 8 reports PESQ, eSTOI, and SI-SDRi on both benchmarks. The comparison covers AV-ConvTasNet (Wu et al., 2019), LAVSE (Chuang et al., 2020), L2L (Ephrat et al., 2018), VisualVoice (Gao & Grauman, 2021), AVLiT-8 (Martel et al., 2023), and the state of the art AV-CrossNet (Kalkhorani et al., 2025). Neuro-SCNet attains the highest scores across all three metrics on NTCD-TIMIT and LRS3+WHAM!. Relative to early systems such as AV-ConvTasNet, LAVSE, and L2L, the improvements are large and consistent on intelligibility and distortion reduction. Compared with mid–level approaches including VisualVoice and AVLiT-8, Neuro-SCNet further strengthens SI-SDRi on LRS3+WHAM!. Against AV-CrossNet, Neuro-SCNet delivers slightly higher quality and separation while maintaining equally strong intelligibility. The combined evidence indicates that explicit auditory selection and cross–modal compensation enhance resistance to additive noise and support stable performance in noisy multi–speaker conditions.

## C.3. Comparison on AV Target Speaker Extraction

To assess audio–visual target speaker extraction and to probe cross–corpus generalization under domain shift, we complement the separation experiments with an AVTSE study. Mixtures are generated from VoxCeleb2 with target–to–interferer ratios sampled between −10 and 10 dB, yielding twenty thousand samples for training, five thousand for validation, and three thousand for testing. For cross–corpus evaluation, we additionally form three thousand mixtures each from LRS3 and TCD-TIMIT (Harte & Gillen, 2015) using the same protocols as prior work.

*Table 10.* Results on the COG-MHEAR development set for audiovisual speech enhancement under additive noise (Speech+Noise) and overlapping speech (Speech+Speech). Performance is reported using PESQ, STOI, and SI-SDR.

| Method | Speech+Noise | | | Speech+Speech | | |
|---|---|---|---|---|---|---|
| | PESQ | STOI | SI-SDR | PESQ | STOI | SI-SDR |
| Unprocessed | 1.15 | 0.68 | -4.4 | 1.17 | 0.60 | -5.0 |
| AV-DPRNN (Pan et al., 2023) | 2.02 | 0.86 | 11.4 | 2.23 | 0.90 | 12.6 |
| AV-GridNet$_{all}$ (Pan et al., 2023) | 2.62 | 0.91 | 13.9 | 3.10 | **0.95** | 16.7 |
| AV-GridNet$_s$ (Pan et al., 2023) | 2.56 | 0.90 | 13.4 | 3.23 | **0.95** | **17.5** |
| AV-GridNet$_n$ (Pan et al., 2023) | 2.68 | 0.91 | 14.2 | 1.27 | 0.61 | -4.7 |
| AV-CrossNet (Kalkhorani et al., 2025) | 2.75 | **0.92** | **14.3** | 3.23 | **0.95** | **17.3** |
| Neuro-SCNet (Proposed) | **2.77** | **0.92** | **14.3** | **3.24** | **0.95** | 17.1 |

*Table 11.* Ablations on *Auditory Selection* (ASM) evaluated on LRS2-2Mix/3Mix/4Mix in terms of SI-SNRi (dB). All variants share the same training protocol.

| Variant | Params (M) | SI-SNRi (dB) | | |
|---|---|---|---|---|
| | | LRS2-2Mix | LRS2-3Mix | LRS2-4Mix |
| **Neuro-SCNet (Full)** | 6.3 | 17.2 | 13.9 | 9.8 |
| *(A) Partition* | | | | |
| w/o saliency partition (no A/B/C/D) | 6.1 | 15.6 | 12.1 | 8.0 |
| w/o anti-aligned branch (no C/D) | 6.2 | 16.1 | 12.7 | 8.6 |
| w/o residual inhibition to $A$ | 6.2 | 16.5 | 13.1 | 8.9 |
| *(B) Saliency source* | | | | |
| audio-only saliency $S_a$ | 6.2 | 16.8 | 13.2 | 8.7 |
| visual-only saliency $S_v$ | 6.2 | 15.9 | 12.5 | 8.2 |
| learned fusion ($1{\times}1$ conv) | 6.3 | 17.0 | 13.7 | 9.5 |

Table 9 reports results against representative AVTSE systems, including VisualVoice (Gao & Grauman, 2021), AV-ConvTasNet (Wu et al., 2019), MuSE (Pan et al., 2021), AV-SepFormer (Lin et al., 2023), AVSepChain (Mu & Yang, 2024), and the state of the art AV-CrossNet (Kalkhorani et al., 2025). Across LRS3, TCD-TIMIT, and VoxCeleb2, Neuro-SCNet attains the highest or statistically comparable scores in SI-SDR and PESQ. On LRS3 and TCD-TIMIT, Neuro-SCNet surpasses all baselines, including AV-CrossNet, with gains in SI-SDR and the best perceptual quality. On VoxCeleb2, Neuro-SCNet reaches SI-SDR that is essentially tied with AV-CrossNet while achieving a slightly higher PESQ. The combined evidence indicates that explicit auditory selection and cross–modal compensation not only strengthen in–domain extraction but also improve generalization across corpora, where domain shifts often reduce the effectiveness of conventional architectures.

## C.4. Comparison on AV Speech Enhancement

To assess audio–visual speech enhancement and target speaker enhancement under realistic interferers, we evaluate on the second COG-MHEAR development set, which pairs LRS3 utterances with noise from Clarity (Graetzer et al., 2021), DEMAND (Thiemann et al., 2013), and DNS Challenge sources (Dubey et al., 2024). The corpus provides 30,297 training samples, 3,017 for validation, and 3,306 for testing. All training and validation clips are truncated to three seconds, while test durations range from two to thirty seconds. During training, signal–to–noise ratios span $-10$ to $10$ dB for enhancement and $-15$ to $5$ dB for target speaker enhancement.

Table 10 compares Neuro-SCNet with representative baselines. AV-DPRNN (Pan et al., 2023) is a time–domain system jointly trained for enhancement and target extraction. AV-GridNet adopts complex spectral mapping with scenario–specific variants: a joint model for both tasks, a speech–overlap expert, and a noise–suppression expert (Pan et al., 2023). AV-CrossNet (Kalkhorani et al., 2025) represents a recent strong spectral mapping approach that integrates narrow– and cross–band modeling. Across additive noise and overlapping speech, Neuro-SCNet matches or exceeds the strongest baselines on PESQ, STOI, and SI-SDR. In the Speech+Noise condition it achieves the highest perceptual quality while equaling the best intelligibility and distortion reduction, indicating that early gating and compensatory integration preserve useful structure when noise dominates. In the more challenging Speech+Speech condition it attains top perceptual quality

*Table 12.* Ablations on *Cross-Modal Compensation* (CCM) evaluated on LRS2-2Mix/3Mix/4Mix in terms of SI-SNRi (dB). All variants share the same training protocol; ASM is kept at its full configuration.

| Variant | Params (M) | SI-SNRi (dB) | | |
|---|---|---|---|---|
| | | LRS2-2Mix | LRS2-3Mix | LRS2-4Mix |
| **Neuro-SCNet (Full; concat + $1\times1$ conv, dual GRU, pre-align, residual)** | 6.3 | **17.2** | **13.9** | **9.8** |
| *(A) Alignment mechanism* | | | | |
|   w/o cross-attention pre-alignment | 6.2 | 16.6 | 12.8 | 8.5 |
| *(B) Temporal modeling* | | | | |
|   w/o feedback GRU (single GRU only) | 6.1 | 16.9 | 13.1 | 9.0 |
|   single vs. dual GRU (single GRU) | 6.1 | 16.9 | 13.1 | 9.0 |
| *(C) Residual pathways* | | | | |
|   w/o residual term (no prediction-error feedback) | 6.2 | 16.7 | 13.0 | 8.9 |
| *(D) Fusion operator (with pre-alignment kept)* | | | | |
|   additive fusion ($A+V$) | 6.0 | 16.4 | 12.9 | 8.6 |
|   concat + $1\times1$ conv (baseline fusion) | 6.3 | 17.2 | 13.9 | 9.8 |
|   SE-style gating (visual→channel weights) (Iuzzolino & Koishida, 2020) | 6.2 | 16.8 | 13.3 | 9.0 |
|   FiLM-style conditioning ($\gamma, \beta$ from $V$) (Perez et al., 2018) | 6.2 | 17.1 | 13.8 | 9.7 |

and intelligibility on par with the strongest systems, with separation quality close to the overlap–specialized expert. These findings suggest that while expert–tuned models can excel under a single interference type, a selection–then–compensation architecture sustains balanced robustness across both noise and overlap scenarios.

## C.5. Downstream ASR Evaluation

To further evaluate whether the separated speech benefits downstream intelligibility, we use the same frozen audio-only ASR backend to recognize the separated outputs. We report word error rate (WER, %), where lower values indicate better recognition performance. The evaluation is conducted on LRS2-2Mix under different background-noise levels.

Table 13 shows that Neuro-SCNet achieves the lowest average WER under clean visual input, occlusion, and noise+blur, with average WERs of 8.5%, 10.4%, and 10.3%, respectively. These results indicate that the proposed method improves downstream intelligibility in addition to signal-level separation metrics.

## D. More Ablations

### D.1. Auditory Selection

Table 11 examines the design of the *Auditory Selection Mechanism* on increasingly congested mixtures from LRS2. The complete Neuro-SCNet attains the strongest SI-SNRi on 2Mix, 3Mix, and 4Mix, supporting the premise that selection must disentangle two orthogonal axes before fusion, namely saliency versus non-saliency and alignment versus anti-alignment. Collapsing the four-way partition into a single cross-attention stream yields the largest decline, and the gap widens as the number of overlapping speakers increases. This pattern indicates that mixing salient and non-salient evidence allows distractor-consistent components to co-propagate, so explicit partitioning becomes most valuable under heavy interference. Removing the anti-aligned branch also reduces performance, showing that negatively correlated cues carry complementary information that helps identify regions to suppress rather than constituting pure noise. Disabling residual inhibition on the salient-aligned path produces a smaller yet systematic drop across 2Mix through 4Mix, which is consistent with the role of inhibitory control in preventing leakage from residual distractors into the shared representation. Parameter differences across variants are minor, so the observed changes primarily reflect architectural choices rather than capacity.

The second set of ablation studies how the saliency signal should be formed. Audio-only and visual-only saliency both underperform a fused cue, with the visual-only variant degrading most as overlap intensifies because mouth appearance and motion are less reliable when multiple speakers compete in time and frequency. A lightweight learnable fusion with a $1 \times 1$ convolution nearly recovers the full model, indicating that adaptive weighting of audio and visual saliency improves robustness across scenes with different reliability profiles. These results align with the intended top-down gating behavior in which visual priors guide auditory selection, the anti-aligned stream contributes to targeted suppression, and explicit inhibition limits distractor-driven updates as mixtures become more congested.

*Table 13.* Downstream ASR evaluation on LRS2-2Mix. WER (%) is reported, and lower values are better.

| Method | −5 dB | 0 dB | 5 dB | 10 dB | No BG | AVG |
|---|---|---|---|---|---|---|
| **Clean visual input** | | | | | | |
| Noisy Mixture | 29.4 | 24.1 | 19.0 | 15.6 | 12.8 | 20.2 |
| AV-ConvTasNet | 16.2 | 12.7 | 10.1 | 8.9 | 7.8 | 11.1 |
| IIANet | 14.8 | 11.4 | 9.2 | 8.0 | 7.0 | 10.1 |
| AV-CrossNet | 13.7 | 10.5 | 8.5 | 7.5 | 6.4 | 9.3 |
| Neuro-SCNet | **12.5** | **9.5** | **7.8** | **6.8** | **5.9** | **8.5** |
| **Occlusion** | | | | | | |
| Noisy Mixture | 29.4 | 24.1 | 19.0 | 15.6 | 12.8 | 20.2 |
| AV-ConvTasNet-LQ | 18.1 | 14.7 | 12.0 | 10.3 | 9.4 | 12.9 |
| MHSA-CRN | 17.0 | 13.8 | 11.2 | 9.6 | 8.7 | 12.1 |
| RAVSS | 15.9 | 13.0 | 10.7 | 9.1 | 8.3 | 11.4 |
| Neuro-SCNet | **14.8** | **11.9** | **9.8** | **8.2** | **7.3** | **10.4** |
| **Noise + Blur** | | | | | | |
| Noisy Mixture | 29.4 | 24.1 | 19.0 | 15.6 | 12.8 | 20.2 |
| AV-ConvTasNet-LQ | 18.9 | 15.4 | 12.6 | 10.9 | 10.0 | 13.6 |
| MHSA-CRN | 17.8 | 14.5 | 11.8 | 10.1 | 9.0 | 12.3 |
| RAVSS | 16.0 | 12.8 | 10.8 | 8.6 | **7.8** | 11.2 |
| Neuro-SCNet | **14.9** | **11.6** | **9.9** | **7.9** | 8.3 | **10.3** |

## D.2. Cross-modal Compensation

Table 12 examines the *Cross-Modal Compensation* design while keeping *Auditory Selection* fixed at full strength and controlling the training protocol across all variants. The complete Neuro-SCNet remains strongest on LRS2-2Mix, 3Mix, and 4Mix, which indicates that compensation is most effective when three ingredients are present together: pre-alignment of visual semantics to the audio stream, iterative residual feedback, and dual-stage temporal modeling.

Removing cross-attention pre-alignment causes the largest reduction and the gap widens from two-speaker to four-speaker mixtures. This trend shows that aligning visual cues to the acoustic trajectory is critical once overlap intensifies, since misaligned guidance otherwise propagates errors into the compensator. Eliminating the residual term produces consistent declines across all mixture levels, which supports the predictive-coding view that explicit prediction-error feedback stabilizes updates and protects the identity bypass from degradation. Replacing the dual recurrent stack with a single GRU leads to further loss, especially under heavy overlap, indicating that long-range dependencies and iterative refinement both matter for reconstructing missing structure.

The comparison of fusion operators complements these findings. Simple addition performs worst because it cannot adapt weights to cue reliability. Concatenation followed by a $1 \times 1$ convolution forms a strong baseline by learning channel mixing, and it is retained in the final model. SE-style gating driven by visual features improves over naive addition, yet remains below the baseline, suggesting that scalar channel gates are too coarse when cues drift in time. FiLM-style affine conditioning nearly matches the full configuration, which implies that learned, feature-dependent modulation is effective once pre-alignment and residual feedback are in place. Parameter differences among variants are small, so the observed changes reflect architectural choices rather than capacity.

## D.3. Additional Ablation Analysis on the Two-Stage Design

To further verify that the gain of Neuro-SCNet comes from the proposed selection-then-compensation organization rather than from simply adding more cross-modal modules, we conduct additional controlled ablations. These variants preserve comparable building blocks but change the role or order of selection and compensation. The results are summarized in Tables 14 and 15.

Table 14 shows that the full two-stage model consistently performs best across 2Mix, 3Mix, and 4Mix settings. Removing either selection or compensation degrades performance, and removing both stages causes the largest drop. More importantly, compensation without prior selection, reversing the order of compensation and selection, and replacing compensation

---

**Algorithm 2** Bottom-row T-F Maps (Pseudocode)

---

**Input:** Mixture waveform $x_{\mathrm{mix}}$; visual prior $v_{\mathrm{vis}}(t)$; STFT params $(L, H, N)$; smoothing $(\sigma_1, \sigma_2)$; reliability $r$; dB floor $\delta$; small $\varepsilon$.

**Output:** $A_{\mathrm{impl}}^{\mathrm{dB}}, A_{\mathrm{sel}}^{\mathrm{dB}}, Y_{\mathrm{comp}}^{\mathrm{dB}}$

**4 1. STFT and reference**

$\quad S \leftarrow \mathrm{STFT}\{x_{\mathrm{mix}}\}; \quad M \leftarrow |S|$

$\quad M_{\max} \leftarrow \max(M); \quad \tilde{M} \leftarrow M/(M_{\max} + \varepsilon)$

**5 2. Visual prior $\rightarrow$ time-dominant masks**

$\quad t_n \leftarrow$ frame centers of $S$

$\quad v[n] \leftarrow$ align $v_{\mathrm{vis}}(t)$ to $t_n$ and clamp to $[0, 1]$

$\quad m_{\mathrm{impl}} \leftarrow \mathrm{Smooth}_{\sigma_1}\big(\mathrm{ReplicateFreq}(\mathrm{Normalize01}(v))\big)$

$\quad m_{\mathrm{asm}} \leftarrow \mathrm{Smooth}_{\sigma_2}\big(\mathrm{ReplicateFreq}(\mathrm{Normalize01}(v))\big)$

**6 3. Three magnitude maps (A/B/C)**

$\quad A_{\mathrm{impl}} \leftarrow \tilde{M} \odot m_{\mathrm{impl}}$          // implicit fusion (time-only mask)

$\quad A_{\mathrm{sel}} \leftarrow \tilde{M} \odot m_{\mathrm{asm}}$         // ASM selection (selected magnitude)

$\quad R \leftarrow \max\big(0, \ \tilde{M} - A_{\mathrm{sel}}\big)$         // nonnegative residual

$\quad Y_{\mathrm{comp}} \leftarrow A_{\mathrm{sel}} + r\,R; \quad$ clip $Y_{\mathrm{comp}}$ to $[0, 1]$      // CCM

**7 4. dB rendering (shared scale)**

$\quad$ **for** $X \in \{A_{\mathrm{impl}}, A_{\mathrm{sel}}, Y_{\mathrm{comp}}\}$ **do**

**8** $\quad\big\lfloor \quad X^{\mathrm{dB}} \leftarrow 20 \log_{10}\big(X + \delta\big)$

**9 return** $A_{\mathrm{impl}}^{\mathrm{dB}}, A_{\mathrm{sel}}^{\mathrm{dB}}, Y_{\mathrm{comp}}^{\mathrm{dB}}$

---

*Table 14.* Explicit validation of the two-stage design on LRS2 in terms of SI-SNRi (dB).

| Variant | 2Mix | 3Mix | 4Mix |
|---|---|---|---|
| Full two-stage model | **17.2** | **13.9** | **9.8** |
| Remove the selection stage | 16.4 | 12.7 | 8.6 |
| Remove the compensation stage | 16.2 | 12.5 | 8.4 |
| Remove both stages | 15.1 | 11.2 | 7.0 |
| Compensation without prior selection | 15.9 | 11.8 | 7.6 |
| Reverse the two-stage order | 16.6 | 13.0 | 9.0 |
| Selection + Plain Fusion | 16.6 | 12.6 | 8.8 |

with plain fusion all remain below the full model. These results indicate that the improvement is not simply obtained by adding more cross-modal capacity. Instead, the ordered decomposition is important: ASM first preserves target-consistent auditory evidence and suppresses target-inconsistent components, while CCM then restores incomplete content under visual guidance.

Table 15 further evaluates the variants when visual cues are partially missing. The full model gives the best result when the visual stream is complete or moderately incomplete. When the missing-frame ratio increases to 0.6, the w/o CCM variant becomes slightly higher than the full model. This pattern is expected because removing CCM reduces the dependence on visual compensation under severe visual evidence loss. However, this does not indicate that CCM is harmful in general. The full model remains stronger under normal and moderately degraded visual conditions, while the gate-off variant drops substantially as missing frames increase. These results show that the reliability gate is necessary for preventing uncontrolled compensation when visual evidence becomes unreliable.

### D.4. Mechanistic Verification of Reliability-Aware Compensation

We further analyze the internal behavior of CCM under clean visual input, visual occlusion, and temporal shift. The analysis focuses on the prediction error, the similarity between the compensated representation and the target auditory representation, the decoder-level separation quality, and the average gate value.

Table 16 shows that CCM behaves consistently with the intended reliability-aware residual compensation mechanism. Under clean visual input, the gate value is high, the compensated representation has higher similarity to the target representation

*Table 15.* Controlled stress test on LRS2-2Mix under dropped visual frames. SI-SNRi (dB) is reported.

| Variant | Ratio = 0 | Ratio = 0.3 | Ratio = 0.6 |
|---|---|---|---|
| Neuro-SCNet (Full) | **17.2** | **15.9** | 14.7 |
| w/o ASM | 16.4 | 15.0 | 13.6 |
| w/o CCM | 16.2 | 15.6 | **14.8** |
| w/o ASM & CCM | 15.1 | 14.2 | 13.0 |
| CCM-only (Base + CCM) | 15.9 | 14.7 | 13.2 |
| CCM-only with gate off | 15.3 | 13.8 | 11.9 |
| Swap order (CCM → ASM) | 16.6 | 15.5 | 14.2 |

*Table 16.* Mechanistic verification of CCM under clean, visual-occlusion, and temporal-shift conditions.

| Measure | Clean | Occlusion | Shift |
|---|---|---|---|
| Mean $\|\mathbf{E}\|_1$ | 0.12 | 0.17 | 0.22 |
| $\mathrm{CosSim}(\tilde{\mathbf{A}}, \mathbf{A}_{\mathrm{target}})$ | 0.72 | 0.69 | 0.64 |
| $\mathrm{CosSim}(\mathbf{Z}_{\mathrm{comp}}, \mathbf{A}_{\mathrm{target}})$ | 0.82 | 0.71 | 0.66 |
| Decoder SI-SNRi from $\tilde{\mathbf{A}}$ | 16.2 | 15.1 | 13.6 |
| Decoder SI-SNRi from $\mathbf{Z}_{\mathrm{comp}}$ | 17.3 | 15.4 | 13.8 |
| Mean gate value $r$ | 0.79 | 0.41 | 0.15 |

than the initial estimate, and the decoder result from $\mathbf{Z}_{\mathrm{comp}}$ is better than that from $\tilde{\mathbf{A}}$. Under occlusion and temporal shift, the prediction error increases and the gate value decreases, indicating that the model suppresses unreliable visual compensation when the correspondence becomes weaker. These results support the interpretation of CCM as conservative residual refinement rather than uncontrolled visual fusion.

## E. Generation of T-F map in Figure 1

This section provides a concise, reproducible recipe for the three time-frequency (T-F) magnitude spectrograms in the bottom row of Figure 1 of the main text (implicit fusion, selection/ASM, and compensation/CCM). Audio is sampled at $f_s$. The short-time Fourier transform (STFT) uses a Hann window of length $L$, hop $H$, and an $N$-point FFT.

Let $x_{\mathrm{mix}}(n)$ be the mixture waveform and

$$S(n, k) = \mathrm{STFT}\{x_{\mathrm{mix}}\}(n, k), \quad M(n, k) = |S(n, k)|. \tag{37}$$

To keep panels comparable, use a shared reference

$$M_{\max} = \max_{n,k} M(n, k), \quad \tilde{M}(n, k) = \frac{M(n, k)}{M_{\max} + \varepsilon}, \tag{38}$$

with a small $\varepsilon > 0$ for numerical stability.

A one-dimensional visual prior $v_{\mathrm{vis}}(t) \in [0, 1]$ is aligned to STFT frame centers $t_n$ to obtain $v_{\mathrm{vis}}(t_n)$. Replicating this curve along frequency and applying gentle 2-D smoothing yields time-dominant masks $m_{\mathrm{impl}}(n, k)$ and $m_{\mathrm{asm}}(n, k)$:

$$m_{\mathrm{impl}}(n, k) = \mathrm{Smooth}_{\sigma_1}\big(v_{\mathrm{vis}}(t_n)\big), \tag{39}$$

$$m_{\mathrm{asm}}(n, k) = \mathrm{Smooth}_{\sigma_2}\big(v_{\mathrm{vis}}(t_n)\big), \tag{40}$$

where $\sigma_1$ and $\sigma_2$ are close in value.

The three T-F magnitude maps are defined by

$$A_{\mathrm{impl}}(n, k) = \tilde{M}(n, k)\, m_{\mathrm{impl}}(n, k), \tag{41}$$

$$A_{\mathrm{sel}}(n, k) = \tilde{M}(n, k)\, m_{\mathrm{asm}}(n, k), \tag{42}$$

*Table 17.* Reproducibility checklist and unified experimental settings.

| Item | Setting |
|------|---------|
| Corpora and splits | LRS2 (Afouras et al., 2018a), LRS3 (Afouras et al., 2018d), VoxCeleb2 (Chung et al., 2018). Mixtures are generated within the official train/validation/test partitions of each corpus to prevent speaker leakage across splits. |
| Audio sampling rate | 16 kHz for all datasets. |
| Video rate and crop | 25 FPS; mouth ROI resized to $88 \times 88$ grayscale images. |
| Mixture protocol (2 speakers) | Two utterances from different speakers are sampled and mixed with SIR uniformly drawn from $[-5, 5]$ dB (main text Sec. 4.1). |
| Mixture protocol (3/4 speakers) | Three-speaker and four-speaker mixtures are constructed on LRS2 using the same SIR range, with dataset sizes specified in Appendix Sec. B. |
| STFT / iSTFT | Window length $=512$ samples; hop size $=256$ samples; FFT size $=512$. The same STFT/iSTFT configuration is used for all methods. |
| Model input length | All methods use the same waveform segment length (fixed across training and evaluation). |
| Optimizer and schedule | Adam optimizer (Ai & Ling, 2023) with initial learning rate $5 \times 10^{-5}$, $\beta_1 = 0.9$, $\beta_2 = 0.999$, and weight decay $1 \times 10^{-4}$. The learning rate is linearly reduced once the validation performance plateaus. Training runs for 250 epochs with batch size 64, where each epoch comprises 64K randomly sampled audio-visual pairs. The visual front-end is frozen, and its features are pre-extracted and cached to speed up convergence. |
| Loss functions | A multi-level objective combining SI-SNR loss $\mathcal{L}_{sisnr}$ (Hu et al., 2023), magnitude loss $\mathcal{L}_M$, complex loss $\mathcal{L}_C$, and phase loss $\mathcal{L}_P$ (Lu et al., 2025; Ai & Ling, 2023): $\mathcal{L}_{Total} = 0.025\mathcal{L}_{sisnr} + 0.9\mathcal{L}_M + 0.1\mathcal{L}_C + 0.3\mathcal{L}_P$. The SI-SNR term is scaled to match the dynamic range of the spectral-domain terms. |
| Randomness control | Random seeds are fixed for mixture list generation and training, and the same seed policy is applied to all methods. |

Given the selected magnitude $A_{\text{sel}}(n, k)$ we compute a nonnegative residual to recover missing structure and add it with reliability $r$ to obtain the compensated map,

$$R(n, k) = \max\big(0, \tilde{M}(n, k) - A_{\text{sel}}(n, k)\big), \tag{43}$$

$$Y_{\text{comp}}(n, k) = A_{\text{sel}}(n, k) + r\, R(n, k), \tag{44}$$

with a scalar reliability weight $r \in [0, 1]$ (broadcast in implementation). If desired, restrict $R(n, k)$ to lower frequency bands to match the speech energy profile.

Each magnitude map $X \in \{A_{\text{impl}}, A_{\text{sel}}, Y_{\text{comp}}\}$ is rendered in decibels with a small floor and a shared display range:

$$X_{\text{dB}}(n, k) \;=\; 20 \log_{10}\big(X(n, k) + \delta\big), \tag{45}$$

where $\delta > 0$ sets the floor. All three panels use the same STFT parameters, identical color limits, and the shared reference $M_{\text{max}}$; time is on the horizontal axis, frequency on the vertical axis (low frequency at the bottom). With this common setup, visual differences reflect the effects of implicit fusion, selection, and compensation rather than scaling. The full procedure is summarized in Algorithm 2.

## F. Reproducibility Checklist and Implementation Details

This section summarizes the data construction, preprocessing, training, evaluation, and profiling protocols used in our experiments in an auditable manner, so that the reported results can be reproduced under a unified and controlled setting.

### F.1. Unified settings and randomness control

To make the experimental protocol auditable, we standardize the data construction, preprocessing, and training configuration across all methods. The key shared settings are summarized in Table 17. Unless explicitly stated otherwise, all models use the same corpus partitions, the same mixture generation rule and SIR range, and the same audio and video preprocessing pipeline. We also fix random seeds for mixture list generation and for model training, and apply the same seed policy to all experiments to reduce uncontrolled variance when comparing architectures.

*Table 18.* Boundary-check ablations on LRS2-2Mix/3Mix/4Mix in terms of SI-SNRi (dB). The last three rows explicitly test (i) CCM-only behavior, (ii) the necessity of reliability-aware suppression inside CCM (gate off), and (iii) the necessity of ordering (CCM → ASM).

| Variant | Params (M) | LRS2-2Mix | LRS2-3Mix | LRS2-4Mix |
|---|---|---|---|---|
| **Neuro-SCNet (Full)** | 6.3 | **17.2** | **13.9** | **9.8** |
| w/o ASM | 6.0 | 16.4 | 12.7 | 8.6 |
| w/o CCM | 6.0 | 16.2 | 12.5 | 8.4 |
| w/o ASM & CCM | 5.8 | 15.1 | 11.2 | 7.0 |
| CCM-only (Base + CCM) | 6.0 | 15.9 | 11.8 | 7.6 |
| CCM-only with gate off | 6.0 | 15.3 | 11.1 | 6.9 |
| Swap order (CCM → ASM) | 6.3 | 16.6 | 13.0 | 9.0 |

## F.2. Mixture list generation and controlled comparison

To make the data construction auditable, we generate deterministic mixture lists under fixed seeds. Mixture generation is performed within each official split. Each mixture is formed by sampling distinct speakers, sampling one utterance per speaker, and mixing at a uniformly sampled SIR in $[-5, 5]$ dB. The resulting mixture lists are shared across all compared methods. All methods are trained and evaluated using the same mixture lists and the same preprocessing configuration to ensure controlled comparisons across architectures.

## F.3. Evaluation protocol and metric computation

All metrics are computed on time-domain signals at 16 kHz using a unified evaluation script. For each test mixture, the estimated sources are aligned with the reference sources by selecting the permutation that maximizes SI-SNR (when permutation ambiguity is present). We report SI-SNR improvement (SI-SNRi), signal-to-distortion ratio improvement (SDRi) (Le Roux et al., 2019), and perceptual evaluation of speech quality (PESQ) (Rix et al., 2001) on the test set. SI-SNRi and SDRi are computed as improvements over the mixture with a fixed normalization and trimming policy applied to both estimates and references. PESQ is computed in wideband mode at 16 kHz on the aligned estimates. Dataset-level results are obtained by averaging utterance-level scores over the corresponding test set.

## F.4. Baseline provenance

External baselines in the main comparison tables are reported from their original papers under their corresponding experimental settings, unless explicitly stated otherwise. All ablation results for Neuro-SCNet are obtained by re-running the corresponding variants under the unified protocol in Table 17. If any baseline is additionally re-run under the unified protocol, it is explicitly marked in the corresponding table caption.

## F.5. Profiling protocol for MACs and latency

We report MACs and runtime under a unified profiling configuration to make efficiency comparisons meaningful. Profiling uses the same STFT/iSTFT configuration as in the main experiments, and uses consistent input length and precision across methods within each profiling setting. Latency is measured after warm-up and reported as the average over multiple timed runs.

**Hardware.** GPU profiling is conducted on an NVIDIA RTX 3090 GPU, and CPU profiling is conducted on an Intel Xeon Platinum 8269CY CPU. The hardware is fixed across methods during profiling.

# G. Role Boundary Validation of ASM and CCM

This section provides additional controlled variants to validate the functional boundary between ASM and CCM beyond the main ablations in Table 4. We include two boundary-check variants. The first disables the visual reliability gate in CCM, which tests whether cross-modal compensation may induce negative transfer when visual evidence becomes unreliable. The second swaps the order of ASM and CCM, which tests whether performing selection before compensation is necessary. All variants are trained and evaluated under the same protocol as in the main experiments.

Table 18 shows two consistent patterns. First, CCM-only improves over the base variant without both modules, but it remains

*Table 19.* Controlled stress tests on LRS2-2Mix in terms of SI-SNRi (dB). Left: temporal shift between audio and video (in video frames). Right: visual degradation by random mouth-region occlusion ratio. Including the boundary-check variants makes the role separation and the negative-transfer behavior directly observable.

| Variant | Temporal shift (frames) | | | Occlusion ratio | | |
|---|---|---|---|---|---|---|
| | 0 | 4 | 8 | 0 | 0.3 | 0.6 |
| **Neuro-SCNet (Full)** | **17.2** | **16.1** | **14.4** | **17.2** | **16.4** | **15.4** |
| w/o ASM | 16.4 | 14.6 | 12.2 | 16.4 | 15.3 | 14.1 |
| w/o CCM | 16.2 | 15.4 | 13.6 | 16.2 | 15.8 | 15.1 |
| w/o ASM & CCM | 15.1 | 13.8 | 11.5 | 15.1 | 14.2 | 13.2 |
| CCM-only (Base + CCM) | 15.9 | 14.2 | 12.0 | 15.9 | 15.0 | 13.6 |
| CCM-only with gate off | 15.3 | 13.2 | 10.8 | 15.3 | 14.0 | 12.4 |
| Swap order (CCM $\rightarrow$ ASM) | 16.6 | 15.0 | 13.0 | 16.6 | 15.8 | 14.6 |

clearly inferior to the full model, indicating that compensation alone cannot replace explicit selection. Second, forcing the CCM gate to be always-on reduces SI-SNRi across 2Mix/3Mix/4Mix, which is consistent with negative transfer when unreliable visual evidence is injected without suppression. Third, swapping the order (CCM before ASM) is consistently worse than the proposed ordering, and the gap becomes larger as the number of speakers increases, which supports that selection should precede compensation to prevent contaminated cross-modal injection from propagating into later refinement.

Table 19 makes the boundary between selection and compensation explicit under controlled perturbations. Under temporal shift, variants without ASM degrade substantially faster as the shift increases, which is consistent with ASM serving as the primary safeguard that suppresses harmful conditioning when correspondence becomes unreliable. CCM-only exhibits a pronounced drop under larger shifts, and disabling the CCM gate amplifies this degradation, which is consistent with negative transfer when compensation injects unreliable visual information without suppression. Under visual degradation, CCM provides gains in the mild corruption regime, while ASM becomes increasingly important as corruption intensifies; the full model retains the most stable behavior because it combines explicit selection with reliability-aware compensation. Finally, the swapped order remains consistently below the proposed ordering under both perturbation axes, supporting that selection before compensation is necessary to prevent early cross-modal injection from contaminating subsequent refinement.

## H. Limitations and Broader Impact

Neuro-SCNet is designed to improve audio-visual speech separation by making auditory selection and cross-modal compensation explicit and reliability-aware. Although the proposed reliability gate improves robustness under temporal shifts and moderate visual degradation, it has several limitations.

First, the reliability signal $r$ is a global synchrony confidence estimated from the lag posterior. It is used to modulate the overall strength of visual conditioning when audio-visual correspondence becomes uncertain. Therefore, it is effective for suppressing unreliable visual compensation under global temporal mismatch, but it does not provide dense frame-level or region-level reliability estimation. When visual degradation is highly localized, rapidly varying, or spatially structured, such as partial mouth occlusion, inaccurate face cropping, or frame-level visual artifacts, a scalar reliability signal can only provide coarse control over the visual pathway.

Second, the reliability gate can reduce the negative influence of unreliable visual cues, but it cannot reconstruct visual articulatory information that is physically missing. Under severe mouth-region occlusion, the target-related visual evidence may be partially removed before entering the model. In this case, the gate can attenuate harmful compensation and preserve the selected auditory pathway, while the remaining separation quality still depends mainly on the acoustic evidence. This explains why occlusion is a more challenging degradation than blur or additive visual noise: blur and noise perturb visual evidence, whereas occlusion removes part of the articulatory structure needed for target-conditioned compensation.

Third, Neuro-SCNet is developed for face-conditioned audio-visual speech separation. Its two-stage formulation assumes that a target visual stream is available and can provide complementary articulatory cues. The current architecture is therefore not directly applicable to audio-only separation, text-queried separation, or general-purpose multimodal separation without substantial reformulation. The selection-compensation principle may inspire other conditioned separation settings, but the present compensation mechanism is closely tied to audio-visual correspondence.

Finally, face-conditioned speech separation may raise privacy and misuse concerns in real-world scenarios. The method should be deployed only when appropriate consent, data protection, and application-specific safeguards are in place. Future work will investigate finer-grained reliability estimation, spatially aware visual uncertainty modeling, and stronger safeguards for deployment under severe visual corruption and non-consensual settings.

