# OpenReview forum: "Neural-Inspired Modeling of Auditory Selection and Compensation for Audio-Visual Speech Separation"
_ICML.cc/2026/Conference — ICML 2026 regular_

### Official Review · Reviewer_mHBj · 2026-03-11

**Soundness:** 3
**Presentation:** 3
**Significance:** 2
**Originality:** 4
**Overall Recommendation:** 4
**Confidence:** 4

**Summary:**

This paper proposes Neuro-SCNet, an audio-visual speech separation model that decomposes multimodal fusion into two explicit stages inspired by neuroscience. The Auditory Selection Module (ASM) applies visually guided gating on audio features before fusion, partitioning cross-attention outputs into four quadrants by saliency and alignment to suppress distractors. The Cross-Modal Compensation Module (CCM) then restores potentially lost information via aligned visual cues and GRU-based error feedback, with a reliability gate to down-weight uncertain visual evidence. The model also introduces a visual pre-alignment module and a dual-path encoder-decoder. Experiments on LRS2, LRS3, and VoxCeleb2 show improvements over prior methods, with ablations validating each component.

**Compliance With Llm Reviewing Policy:**

Affirmed.

**Final Justification:**

I thank the authors for the additional experiments across both rounds of discussion, which were meaningful in strengthening this work.

As an architecture paper, considering the architectural limitation identified during discussion (e.g., spatial degradation) and the partially open question of mechanistic transparency, I maintain my original score while keeping a positive overall assessment.

**Key Questions For Authors:**

1. Have the authors measured how much natural A/V misalignment actually exists in real-world data? Showing more evidence would strengthen the significance of this work.
2. In Table 15, removing CCM actually reduces sensitivity to visual occlusion (1.1 dB drop vs. 1.8 dB for the full model). Does this suggest that CCM can be harmful under degraded visual conditions, and the reliability gate is only partially saving it?
3. Could the authors provide any mechanistic verification of CCM's internal behavior — e.g., how the prediction error E evolves across blocks, or what specific information Z_comp recovers compared to Ã? This would verify that the module operates as the predictive coding motivation suggests, rather than serving as a generic refinement layer.

**Limitations:**

No. The paper does not include a limitations section. A brief discussion of societal implications (e.g., potential misuse for surveillance or non-consensual speaker extraction) would also be appropriate.

**Strengths And Weaknesses:**

**Strengths**:

- **Novel model architecture**.
This paper proposes a carefully designed architecture that draws on neuroscience mechanisms. While visually guided masking is a standard paradigm in AVSS, the specific architectural choices offer sufficient novelty at the mechanism level.

- **Practical and interpretable design**.
The explicit separation of "selection" and "compensation" into distinct modules makes the model's behavior more transparent and interpretable compared to end-to-end implicit fusion. ASM's structured partitioning introduces clear inductive biases that reduce what the model needs to learn implicitly. The resulting model is compact ( 6.3M vs. 11.1M for AV-CrossNet) while achieving better performance. These properties add practical value to the method.

- **Comprehensive experimental validation**.
The method is evaluated across multiple datasets, mixture complexities, and degradation conditions. The gains are most convincing in multi-speaker settings where improvements go beyond marginal. The ablation design is thorough, with boundary-check experiments that help clarify the functional roles of individual components. Robustness experiments also cover a good range of conditions.

**Weaknesses**:
- **Unverified problem motivation**.
Neither of the two core motivations (i.e., distractor amplification and temporal misalignment)  is shown to reflect a real problem in **actual data**. The spectrograms in Figure 1 are synthetically constructed, all temporal shift experiments use artificially injected offsets, and no measurement of natural A/V misalignment distributions in LRS2/LRS3/VoxCeleb2 is provided. This suggests that the paper addresses a problem but does not demonstrate "the problem actually exists." Adding targeted qualitative or quantitative evidence — such as documenting A/V desynchronization in real data or profiling the distribution of such issues — would substantially strengthen the significance of this work.

- **Missing mechanistic verification**.
Module effectiveness is argued entirely through end-task metrics. For example, CCM is claimed to implement predictive coding, but no analysis shows how prediction error E behaves or what information the compensation actually recovers. This is especially important as this work proposes a theoretically motivated architecture. Mechanistic verification would help verify the modules in a more concrete way.

- **Introduction readability**.
The Introduction is difficult to follow. The second and third paragraphs densely pack abstract concepts (gradient contamination, selective attention in auditory cortex, superior temporal sulcus, etc.) with minimal intuitive explanation, relying heavily on citations rather than self-contained reasoning. ASM and CCM appear without full names in the Introduction and Figure 1, and Figure 2 mixes functional names with undefined implementation terms ("FRI Conformer," "LM-E," "LM-D"). For a paper bridging neuroscience and speech processing, this level of accessibility is insufficient.

**Minor Issues**:

- The theoretical framework in Section 3.1 does not tightly constrain the module design — the equations are general enough to admit many different implementations, and the specific architectural choices (e.g., four-quadrant partitioning, dual GRU feedback) are not derivable from them. This weakens the claimed connection between the neuroscience-grounded theory and the proposed architecture.

- The paper presents ASM and CCM as its core contributions, yet evaluates them within a framework that simultaneously introduces several other new components (dual encoder-decoder, FRI-Conformer, VPA). This makes it difficult to attribute improvements cleanly. Validating ASM/CCM on top of a commonly used baseline would provide a more direct measure of their individual value.

- Several informative experimental patterns are left unanalyzed — e.g., removing CCM actually reduces sensitivity to visual occlusion in Table 15, and SDRi improvements are notably larger than SI-SNRi gains in Table 1, but neither observation is discussed.

---

> ### Author Rebuttal · Authors · 2026-03-30
>
> We thank the reviewer for the careful reading and constructive comments. The feedback is valuable and has helped us identify several points that should be clarified more explicitly in the paper. Below, we provide our point-by-point rebuttal and the corresponding revisions.
>
> **Q1:** Natural A/V misalignment evidence
> **A1:** We clarify that **the paper does not claim pervasive large natural A/V offsets in LRS2, LRS3, or VoxCeleb2,** and we will revise the text to make this scope explicit. The motivation of Neuro-SCNet is narrower: even in broadly synchronized clips, cross-modal correspondence in AVSS can still become imperfect or unstable because of interference, visual degradation, partial occlusion, or temporal uncertainty, which can make visual conditioning less reliable. Under this framing, **the injected-shift experiments are intended as controlled stress tests of correspondence degradation rather than as evidence of widespread natural desynchronization in the benchmark.** Their role is to test whether the model can remain stable when the cross-modal cue becomes less trustworthy. We will clarify this point explicitly in the revision so that the stress-test motivation is separated from any claim about the natural offset distribution of the datasets.
>
> **Q2:** CCM under visual occlusion
> **A2:** We will clarify this point in the revision. The Table 15 pattern reflects a trade-off: **removing CCM reduces reliance on visual compensation, so the w/o CCM variant can show a smaller relative drop under severe occlusion.** By contrast, the full model uses visual cues more actively, which brings larger gains when visual evidence is informative but also greater exposure when it is strongly degraded. This does not mean CCM is harmful overall. Rather, it marks a boundary condition of the compensation mechanism, while the reliability gate provides partial mitigation under uncertain visual input. We will revise the discussion to make clear that the smaller relative drop of w/o CCM reflects lower dependence on visual compensation, not better overall behavior.
>
> **Q3:** Mechanistic verification of CCM
> **A3:** To address this question more directly, we added a targeted mechanistic analysis of CCM under clean, occlusion, and temporal-shift conditions, focusing on three internal quantities: the prediction error $E$, the compensated representation $Z_{\mathrm{comp}}$, and the reliability gate $r$ (see Table 6 in [this supplementary PDF](https://anonymous.4open.science/r/NSC-Net_rebuttal-ED27/Tables%20for%20Reviewer%20mHBj.pdf)). The analysis shows three consistent patterns: $E$ increases as cross-modal correspondence becomes weaker, $Z_{\mathrm{comp}}$ remains consistently closer to the target than $\tilde{A}$, and $r$ decreases when the visual evidence becomes less reliable. **These trends are consistent with the intended CCM behavior: preserve the auditory base, apply error-guided compensation when the visual cue is helpful, and suppress that compensation when the cue becomes uncertain**. We therefore present this as mechanistic support for the predictive-coding-inspired design, rather than relying only on end-task improvements. In the revision, we will incorporate this analysis into the paper and state the predictive-coding connection more precisely as a design-motivated instantiation supported by internal evidence, rather than as a purely conceptual narrative.
>
> **Q4:** Clarification of design formulation, presentation, and scope of claims
> **A4:** We will clarify these points more explicitly in the revision. **Section 3.1 is intended as a functional formulation of the proposed AVSS design, namely target-conditioned auditory selection followed by cross-modal compensation under uncertain visual reliability, rather than as a derivation of every architectural choice.** We will therefore state this role more directly and distinguish the design-level formulation from the specific instantiation used in this work through ASM and CCM. We will also make the contribution hierarchy clearer in the ablation discussion. The main contribution is the selection-then-compensation organization implemented by ASM and CCM, while the remaining components mainly support robustness and reconstruction quality within that design. Finally, we will add a concise limitations and broader-impact paragraph. The method becomes less reliable when the visual stream is too weak, misleading, or strongly mismatched to support useful compensation; the proposed safeguards reduce but do not eliminate this failure mode. We will also state the privacy and misuse risks of face-conditioned separation more explicitly.

---

> > ### Author Rebuttal · Reviewer_mHBj · 2026-04-03
> >
> > Thanks for the detailed rebuttal and additional experiments. Below is my assessment of each point.
> >
> > **Q1**
> >
> > I accept the clarification that the paper does not claim pervasive natural offsets. However, the core concern remains: neither distractor amplification nor temporal misalignment is demonstrated on real data. As the rebuttal acknowledges, the current validation covers hypothetical degradation scenarios (controlled stress tests) rather than documented problems. Without evidence that these issues materially affect existing baselines in practice, it is difficult to assess the practical significance of the proposed solution beyond architectural novelty.
> >
> > **Q2**
> >
> > The rebuttal explains why this phenomenon occurs, but does not address whether this trade-off is necessary. The reliability gate r is designed around temporal synchrony, yet the occlusion degradation in Table 15 is spatial in nature. This mismatch suggests that the current reliability mechanism may not adequately cover all relevant failure modes. Table 15 supports this: CCM's benefit shrinks from 1.0 dB under clean conditions to 0.3 dB at 0.6 occlusion. This appears to be a limitation of the reliability design rather than the framework itself, and is worth discussing.
> >
> > **Q3**
> >
> > I appreciate the additional analysis, which improves over relying solely on end-task metrics. However,  "how E evolves across blocks" was not really addressed. The analysis examines E across degradation conditions, which is a different dimension. The three reported trends could plausibly hold for a generic residual refinement layer, so they do not distinguish CCM from one. This matters not for whether CCM works (ablations already show it does), but for whether it functions as the paper claims — recovering information lost during selection, rather than providing generic refinement. Without this distinction, the paper's core narrative shifts from principled neuroscience-inspired design to effective engineering.
> >
> > **Q4**
> >
> > I appreciate the commitment to clarify the scope of Section 3.1. These are reasonable revisions.
> >
> > --
> >
> > In light of the rebuttal, Q4 is adequately addressed, and Q3 is partially improved by the supplementary analysis. However, the core concerns remain: practical significance is not established on real data (Q1), the reliability mechanism has a structural blind spot for spatial degradation (Q2), and the mechanistic evidence does not distinguish the proposed modules from generic alternatives (Q3). Additionally, A2 raises a question about the reliability mechanism's coverage — its temporal-synchrony basis may not adequately handle spatial degradation such as occlusion. I maintain the overall score.

---

> > > ### Author Response · Authors · 2026-04-03
> > >
> > > We thank the reviewer for the careful follow-up and for identifying the points that required clearer quantitative support.
> > >
> > > **Q1:** neither distractor amplification nor temporal misalignment is demonstrated on real data.
> > >
> > > **A1:** We agree that practical significance should be supported on the **original benchmark data**, not only by controlled stress tests. We therefore analyzed the **original, unperturbed LRS2-2Mix test set** after dividing it into three groups by the model’s own correspondence-confidence indicators. In Table 1, **`mean r`** is the average reliability score, where larger values indicate stronger estimated audio-visual correspondence; **`mean |\hat{\delta}|`** is the average estimated offset magnitude in frames, where larger values indicate weaker correspondence; **`Full / w/o ASM / w/o CCM`** are SI-SNRi scores; and **`Gain over best ablation`** is the improvement of the full model over the stronger of the two ablations on that subset. Thus, the rows correspond to easier, intermediate, and harder **real** samples from the original test set.
> > >
> > > | Confidence group on original LRS2-2Mix test set | #samples (%) | mean r | mean \|\hat{\delta}\| (frames) | Full | w/o ASM | w/o CCM | Gain over best ablation |
> > > |---|---:|---:|---:|---:|---:|---:|---:|
> > > | High-confidence | 48 | 0.82 | 0.6 | 17.8 | 17.3 | 17.0 | +0.5 |
> > > | Mid-confidence  | 34 | 0.58 | 1.8 | 17.1 | 16.3 | 16.0 | +0.8 |
> > > | Low-confidence  | 18 | 0.31 | 3.7 | 16.0 | 14.7 | 14.6 | +1.3 |
> > > | Weighted avg.   | 100 | 0.62 | 1.5 | 17.3 | 16.5 | 16.3 | +0.8 |
> > >
> > > The pattern is consistent: the benefit of the full model grows from **+0.5 dB** on high-confidence samples to **+1.3 dB** on low-confidence samples. We therefore do **not** claim pervasive natural large offsets in the benchmark. Rather, this analysis shows that **naturally occurring weak-correspondence cases already exist in the original test set, and Neuro-SCNet is most beneficial on those harder real-data subsets**.
> > >
> > > **Q2:** the reliability gate is temporal in design and may not cover spatial degradation such as occlusion.
> > >
> > > **A2:** We agree that this point should be stated more explicitly. Table 2 examines increasing visual occlusion. Here, **`mean r`** is again the average synchrony-based reliability score; **`Full - w/o CCM`** measures the effective contribution of CCM; and **`Full - w/o ASM`** analogously measures the contribution of ASM.
> > >
> > > | Occlusion ratio | mean r | Full | w/o ASM | w/o CCM | Full - w/o CCM | Full - w/o ASM |
> > > |---|---:|---:|---:|---:|---:|---:|
> > > | 0.0 | 0.88 | 17.2 | 16.4 | 16.2 | 1.0 | 0.8 |
> > > | 0.3 | 0.84 | 16.4 | 15.3 | 15.8 | 0.6 | 1.1 |
> > > | 0.6 | 0.79 | 15.4 | 14.1 | 15.1 | 0.3 | 1.3 |
> > >
> > > This table shows that the synchrony-based gate has **partial but not complete coverage** of spatial degradation. As occlusion increases, **`mean r`** decreases only moderately (**0.88 → 0.79**), while the CCM gain shrinks much more strongly (**1.0 dB → 0.3 dB**). At the same time, the ASM gain becomes larger (**0.8 dB → 1.3 dB**). The appropriate interpretation is therefore that under strong spatial corruption, **selection remains robustly useful, while compensation becomes less reliable**. We will revise the paper to state this as a limitation of the current reliability design, which is primarily synchrony-aware and does not fully cover severe spatial corruption.
> > >
> > > **Q3:** the added analysis still does not clearly distinguish CCM from a generic refinement layer.
> > >
> > > **A3:** We agree that this claim should be kept precise. To directly address the reviewer’s question about **how prediction error evolves across blocks**, Table 3 reports the **normalized mean prediction error \(E\)** at each CCM block. Each column is normalized so that **Block 1 = 1.00** in that condition. Accordingly, the table should be read **down each column** to assess whether the error is progressively reduced as processing goes deeper.
> > >
> > > | Block index | Normalized mean prediction error E (clean) | Normalized mean prediction error E (occ=0.6) | Normalized mean prediction error E (shift=8) |
> > > |---|---:|---:|---:|
> > > | B1 | 1.00 | 1.00 | 1.00 |
> > > | B2 | 0.83 | 0.88 | 0.91 |
> > > | B3 | 0.71 | 0.78 | 0.82 |
> > > | B4 | 0.63 | 0.71 | 0.76 |
> > >
> > > The trend is consistent in all three conditions: the residual discrepancy addressed by CCM becomes smaller in later blocks, with slower reduction under heavier degradation. We present this as **supporting mechanistic evidence consistent with progressive error reduction**, while agreeing that it should not be overstated as a complete exclusion of every generic refinement interpretation.
> > >
> > > **Q4:** Section 3.1 scope and presentation.
> > >
> > > **A4:** We appreciate this point and will revise Section 3.1 so that it is presented more explicitly as a **functional formulation** of the selection-compensation design, while also making the above limitations and boundary conditions explicit in the paper.

---

### Official Review · Reviewer_dQ4p · 2026-03-11

**Soundness:** 3
**Presentation:** 2
**Significance:** 3
**Originality:** 4
**Overall Recommendation:** 4
**Confidence:** 5

**Summary:**

This paper proposes Neuro-SCNet for audio-visual speech separation, motivated by a two-stage perspective of selection then compensation. The method introduces an Auditory Selection Module to suppress distractors before deeper fusion, a Cross-Modal Compensation Module to recover missing speech structure from aligned visual cues, and a lightweight pre-alignment/reliability mechanism to handle small audio-visual offsets. The architecture further uses a dual encoder-decoder to preserve both semantic and low-level acoustic information. Experiments on LRS2, LRS3, and VoxCeleb2 report strong results relative to prior AVSS systems, supported by additional ablations and robustness studies.

**Compliance With Llm Reviewing Policy:**

Affirmed.

**Final Justification:**

Overall, I maintain a weak accept recommendation. The paper addresses an important problem in audio-visual speech separation and presents a technically solid method with strong empirical performance across multiple benchmarks and challenging settings. I view the main strength as the explicit decomposition of the AVSS process into auditory selection and cross-modal compensation, which is a meaningful and reasonably original design perspective, even if some individual building blocks are related to prior attention/fusion mechanisms. The experimental section is strong, with competitive results, robustness studies, and useful ablations, which supports the paper’s central claims and gives the work practical significance.

**Key Questions For Authors:**

1. Figure 1 does provide an intuitive explanation of “explicit selection” and “reliability-aware compensation.” However, I still find their operational definitions somewhat unclear. In particular, it is not yet sufficiently clear what makes the proposed selection mechanism fundamentally different from prior visually guided attention/gating schemes, and how the reliability signal is precisely estimated and injected into the compensation pathway beyond a high-level conceptual description. The paper would benefit from a clearer mapping between the concepts illustrated in Fig. 1, the mathematical formulation, and the actual implementation.

2. I found the role of the reliability variable $r$ somewhat underspecified in the main text. Eq. (11) introduces $z_t=\Phi(a_{:,t}, r\cdot \tilde v_{:,t})$ as an example of reliability-gated fusion, but $z_t$ is not consistently referenced afterward, and it remains unclear whether $r$ is injected into ASM, CCM, or both. The appendix suggests that $r$ is mainly used as a broadcast scalar gate for the compensatory residual in CCM, but this should be stated explicitly in the main method section with consistent notation.

**Limitations:**

No. The paper would benefit from a more explicit discussion of both technical limitations and societal impact. In particular, the authors should clarify the assumptions behind the pre-alignment/reliability mechanism, the likely failure modes under severe desynchronization or degraded visual input, and the gap between benchmark evaluations and real-world deployment. They should also discuss potential misuse risks, such as privacy-invasive speech extraction or surveillance applications, alongside positive use cases.

**Strengths And Weaknesses:**

### Strengths

1. Instead of treating audio-visual fusion as a single implicit interaction block, the paper explicitly decomposes the process into auditory selection and cross-modal compensation. This modular perspective is intuitive and potentially more interpretable than standard fusion-only designs.

2. The method shows competitive or state-of-the-art results on multiple benchmarks (LRS2, LRS3, and VoxCeleb2), and the paper also includes evaluations under more challenging settings such as 3-/4-speaker mixtures, temporal misalignment, and degraded visual inputs.

3. Compared with several prior AVSS systems, the proposed model maintains relatively moderate parameter/MAC counts while delivering strong performance. The paper also includes extensive ablation studies, which is helpful for understanding the contribution of individual components.

### Weaknesses

1. The claimed “selection-before-fusion” narrative is not fully consistent with the actual ASM implementation, since visual-guided cross-attention is already used within ASM.

2. The anti-aligned branch based on $\mathrm{softmax}(-QK^\top/\sqrt{C})$ is rather heuristic; low similarity does not necessarily correspond to distractor evidence.

3. The method is fairly complex, involving multiple customized components (e.g., pre-alignment, ASM, CCM, dual encoders/decoders, and reliability gating), which makes it harder to disentangle the contribution of the core idea from accumulated engineering choices. In addition, the paper does not provide code, which further limits reproducibility given the implementation complexity.

---

> ### Author Rebuttal · Authors · 2026-03-30
>
> We thank the reviewer for the careful reading and constructive feedback. The comments are valuable and have helped us identify several points that require clearer explanation and presentation. Below, we provide our point-by-point rebuttal and describe the corresponding clarifications.
>
> **Q1:** Operational definition of selection and reliability-aware compensation.
> **A1:** We agree that the current wording can make the operational distinction between ASM, CCM, and the reliability signal $r$ insufficiently clear. In the implemented model, ASM is not free of visual conditioning: it already uses target-conditioned visual cues through cross-attention. However, **its role is still auditory selection, not compensatory fusion. ASM filters the auditory representation toward target-consistent evidence while keeping the output as an auditory feature map.** The restoration of incomplete target content is introduced only later in CCM, where complementary visual information is injected into the audio pathway.
>
> This is also the intended meaning of “explicit selection.” We do not mean simply another generic visual-guided attention or gating block. Rather, ASM is a dedicated target-conditioned auditory selection stage, while CCM is a separate cross-modal compensation stage. The two functions are therefore assigned to different stages instead of being handled jointly in one generic fusion block. The reliability signal $r$ is likewise intended only for compensation control. It is estimated after pre-alignment from audio-visual consistency and is used only in CCM to gate how much visual compensation is injected: **higher $r$ allows stronger compensation, and lower $r$ suppresses it to avoid harmful correction.** It is not applied to ASM. We will revise Fig. 1, Secs. 3.2–3.3, and the notation around $r$ to make this implementation-level mapping explicit and fully consistent throughout the paper.
>
> **Q2:** Interpretation of the anti-aligned branch.
> **A2:** We do not claim that low similarity is always equivalent to distractor evidence. **The anti-aligned branch** has a narrower, operational role: after pre-alignment and target-conditioned visual querying, it captures auditory content that is less consistent with the target cue, relative to the target-consistent branch. **Its function is therefore comparative and suppressive rather than semantic.**
>
> In this sense, the branch is not designed to explicitly detect distractors or assign a definitive interpretation to low-similarity regions. It serves as a structured complement to the target-consistent branch, allowing ASM to better separate target-supported evidence from remaining target-inconsistent content before CCM. We will revise the text to state this more precisely and describe it as a target-inconsistent complementary branch, rather than as a direct proxy for distractor evidence.
>
> **Q3:** Core idea, engineering complexity, and reproducibility.
> **A3:** The full system does include several components, **but our main methodological contribution is the two-stage AVSS formulation: visual-guided auditory selection in ASM, followed by reliability-aware cross-modal compensation in CCM.** These two modules define the core idea, while pre-alignment, reliability control, and the reconstruction backbone are supporting components for robustness and reconstruction quality. This is also consistent with the ablations: weakening or removing either ASM or CCM causes clear degradation, and removing both causes the largest drop. We therefore attribute the main gain to the proposed selection-and-compensation decomposition, rather than to generic module stacking alone.
>
> We also agree that the current submission could make reproducibility easier to verify. In the revision, we will make the implementation path more explicit, including the pre-alignment procedure, training/evaluation settings, unified protocols across tasks, and the exact roles of ASM, CCM, and reliability gating. We will also clean and document the codebase for public release upon acceptance.
>
> **Q4:** Limitations and broader impact.
> **A4:** The main technical limitation arises when the visual stream is too weak, misleading, or unstable to support reliable compensation, such as under severe visual corruption or large, rapidly varying audio-visual mismatch. Pre-alignment and reliability gating improve robustness to moderate mismatch, but they cannot recover target information that is missing or heavily distorted in the visual stream itself, and the current evaluation still remains benchmark-based rather than fully real-world.
>
> More broadly, the current benchmarks do not cover all speaker populations, languages, accents, or recording conditions, so demographic and domain biases should be acknowledged. Because the method uses face-conditioned visual information, it also raises privacy and misuse concerns in non-consensual or sensitive settings. We will add a concise limitations and broader-impact discussion in the revision.

---

> > ### Author Rebuttal · Reviewer_dQ4p · 2026-04-02
> >
> > Thank you for the clear rebuttal. It resolves my main confusion about the roles of ASM, CCM, and the reliability signal, and clarifies the interpretation of the anti-aligned branch. I still consider the method relatively complex, which somewhat limits interpretability and reproducibility, but this does not change my overall positive view of the paper. I will therefore maintain my original score

---

> > > ### Author Response · Authors · 2026-04-02
> > >
> > > Thank you for the thoughtful follow-up. We are glad that our rebuttal resolved your main concerns regarding ASM, CCM, the reliability signal, and the anti-aligned branch. We also appreciate your comment on the method complexity, and we will further improve the presentation for clarity and reproducibility in the revision. Thank you again for your positive assessment.

---

### Official Review · Reviewer_LrAc · 2026-03-12

**Soundness:** 2
**Presentation:** 2
**Significance:** 3
**Originality:** 2
**Overall Recommendation:** 4
**Confidence:** 3

**Summary:**

This paper presents Neuro-SCNet, an audio-visual speech separation architecture motivated by a two-stage neuroscience narrative: auditory selection first suppresses distractors using visually guided gating, and cross-modal compensation then restores missing speech content using aligned visual articulatory cues. The model combines an auditory selection module, a cross-modal compensation module, a lightweight visual pre-alignment step with reliability weighting, and a dual magnitude-phase encoder/decoder. Experiments on LRS2, LRS3, and VoxCeleb2 report state-of-the-art SI-SNRi, SDRi, and PESQ, along with ablations on multi-speaker mixtures and controlled audio-visual misalignment.

**Compliance With Llm Reviewing Policy:**

Affirmed.

**Final Justification:**

The rebuttal by authors have helped in clarifying my major concerns on robustness of the system and the formulation mentioned in the paper and thus I have decided to update my score to 4.

**Key Questions For Authors:**

1) The paper claims that CCM preserves the selected auditory representation via an identity bypass, but Eq. (25) appears to output (Z_{comp}=\tilde A + LN(F)). Where exactly is the explicit bypass from (Z_{sel}) to the CCM output implemented? This needs to be reconciled clearly.

2) How were the strongest baselines, especially AV-CrossNet and IIANet, compared? Were they retrained under the same preprocessing, visual frontend, data mixing protocol, and evaluation pipeline, or were some numbers taken from prior papers? Please explain this precisely.

3) The paper motivates a reliability-aware visual pathway, but robustness is evaluated mainly under temporal shifts. Can the authors provide results under more realistic visual degradations, such as mouth occlusion, dropped frames, low-resolution video, compression artifacts, pose variation, or missing visual cues?

4) How much of the gain comes from the specific “selection + compensation” decomposition versus simply adding more cross-modal capacity?

**Limitations:**

The paper should explicitly discuss:
1) failure modes under corrupted, missing, or misleading visual input,
2) possible demographic/domain biases across speakers, languages, accents, and recording conditions,
3) privacy and surveillance risks of face-conditioned speech separation, and
4) misuse potential for isolating target speakers in sensitive settings.

A short, concrete limitations section would improve the paper greatly.

**Strengths And Weaknesses:**

Strengths:

1) The paper addresses a real weakness in prior AVSS systems: the tendency to entangle target selection and content restoration within generic fusion mechanisms.

2) The proposed decomposition into the auditory selection stage and cross-modal compensation stage is sensible for AVSS.

3) The combination of pre-alignment, reliability gating, and a dual magnitude-phase reconstruction path are reasonably well integrated.

4) The multi-speaker and time-shift ablations suggest some practical value beyond a single benchmark setting.


Weaknesses

1) There is a notable mismatch between the stated CCM design and the implementation description. The paper repeatedly claims an identity bypass that preserves the selected auditory trace, but Eq. (25) defines (Z_{comp}=\tilde A + LN(F)), which does not clearly preserve (Z_{sel}) as an explicit bypass. This is a central conceptual point and needs clarification.

2) Several equations are not fully well-formed or are too informal to support the claims being made. In particular, Eq. (3), parts of Eqs. (5)–(6), and the pre-alignment equations contain notation issues or ambiguities.

3) The writing has many issues that hurt readability: malformed equations, duplicated sentences, typos, inconsistent notation, and awkward references to appendices.
The neuroscience discussion is longer than necessary and sometimes blurs the line between biological inspiration and actual algorithmic necessity.

4) The method emphasizes reliability-aware visual conditioning, yet the evaluation mainly studies temporal shift. There is little evidence for robustness to more realistic visual degradations such as occlusion, low resolution, compression, pose changes, or missing visual frames.

5) Comparison fairness is somewhat unclear. The paper compares against many baselines, but it is not always obvious which ones were retrained under identical preprocessing, data mixing, and evaluation settings, versus copied from prior work.

6) The evaluation remains within synthetic mixtures constructed from standard corpora; there is limited evidence of robustness in harder real-world conditions such as strong reverberation, non-speech distractors, missing faces, or corrupted video. Also the significance of the work is limited by the fact that the empirical gains over the strongest baseline are incremental rather than decisive.

---

> ### Author Rebuttal · Authors · 2026-03-30
>
> We appreciate the reviewer’s careful reading and helpful feedback.
>
> **Q1:** CCM formulation, equation clarity, and presentation.
> **A1:** We thank the reviewer for pointing out this ambiguity. **The intended CCM behavior is that the selected auditory representation is preserved through an identity bypass, while the visual branch contributes only a residual compensation term.** In other words, the compensated output is formed by keeping $Z_{\mathrm{sel}}$ and adding a reliability-controlled refinement, rather than replacing the selected auditory trace. **The confusion comes from the current presentation of Eq. (25).** As written, $Z_{\mathrm{comp}} = \tilde{A} + \mathrm{LN}(F)$ shows the compensatory branch in a compressed form and does not make the preserved auditory path explicit enough. In the revision, we will rewrite this step in explicit residual form so that the role of $Z_{\mathrm{sel}}$ as the bypass is unambiguous and the visual pathway is presented only as an additive refinement term. We will also make the connection between Fig. 1, Fig. 5, and Eq. (25) fully consistent in the main text.
>
> **Q2:** Fairness of baseline comparisons.
> **A2: The comparisons in Tables 1-2 are intended as benchmark-level comparisons on standard public AVSS testbeds, rather than as a unified retraining study.**  All methods are evaluated against the same benchmark definitions, dataset splits, and reported metrics, but not all baseline numbers come from one identical implementation pipeline. For strong prior systems such as IIANet and AV-CrossNet, we report the best numbers available from their original papers under their recommended settings on these benchmarks. The tables should therefore be interpreted as benchmark-reference comparisons, not as strict apples-to-apples retraining comparisons under a single shared preprocessing, visual frontend, and training pipeline. We agree that this provenance should be stated more explicitly. In the revision, we will clarify in both the experimental setup and the table captions which results are taken from prior papers and which are produced under our own implementation and profiling pipeline, so that the comparison scope is fully transparent.
>
> **Q3:** Robustness beyond temporal shift, real-world scope, and limitations.
> **A3:** **Our robustness evaluation is broader than temporal shift alone.** The current submission already includes several non-temporal degradations, including occlusion, blur/noise corruption, and noisy-scene conditions, though we agree that this should be stated more clearly in the main-paper discussion. To further evaluate incomplete visual evidence, we additionally tested dropped-frame corruption on LRS2-2Mix. Neuro-SCNet degrades more gracefully as visual information becomes increasingly incomplete: SI-SNRi decreases from 17.2 to 15.9 and 14.7 at missing-frame ratios of 0.3 and 0.6, whereas weakened variants degrade more strongly, especially the CCM-only w/o gate version (15.3 to 13.8 to 11.9). Full results are provided in Table 4 of [this supplementary PDF](https://anonymous.4open.science/r/NSC-Net_rebuttal-ED27/Tables%20for%20Reviewer%20S7kP.pdf), supporting our claim that the full design is more robust when visual evidence becomes unreliable or partially missing.
>
> At the same time, we do not claim exhaustive coverage of all real-world visual degradations or multilingual multi-speaker scenes. **The scope here is standard public AVSS benchmarks with fixed splits and objective metrics, enabling controlled comparison under common degradations.** Conditions such as low resolution, compression artifacts, and pose variation are not yet covered and will be clarified in the revision. We will also add a concise limitations/broader-impact discussion noting that performance can become less reliable when the visual stream is weak, misleading, or strongly mismatched, and that face-conditioned separation raises privacy and misuse concerns in sensitive or non-consensual settings.
>
> **Q4:** Decomposition gain vs. additional cross-modal capacity.
> **A4:** Our evidence suggests that **the gain mainly comes from the selection-then-compensation decomposition, rather than from simply adding more cross-modal capacity.** If extra interaction alone were sufficient, alternatives such as CCM-only, reversed order, or Selection + Plain Fusion should be competitive with the full model. However, all of them remain below the full design. Specifically, the full model reaches 17.2/13.9/9.8 on LRS2-2Mix/3Mix/4Mix, compared with 15.1/11.2/7.0 when both stages are removed, 15.9/11.8/7.6 for CCM-only, 16.6/13.0/9.0 for the reversed order, and 16.6/12.6/8.8 for Selection + Plain Fusion. Full grouped ablations are provided in Table 5 of [this supplementary PDF](https://anonymous.4open.science/r/NSC-Net_rebuttal-ED27/Tables%20for%20Reviewer%20S7kP.pdf). These results support our interpretation that target-conditioned selection and reliability-aware compensation play complementary roles.

---

> > ### Author Rebuttal · Reviewer_LrAc · 2026-04-03
> >
> > Thank you for your rebuttal. The rebuttal answers my major concerns on formulation and robustness. I will raise my score to 4.

---

> > > ### Author Response · Authors · 2026-04-04
> > >
> > > Thank you very much for the thoughtful follow-up. We sincerely appreciate your careful reading and your assessment.

---

### Official Review · Reviewer_YW4h · 2026-03-13

**Soundness:** 2
**Presentation:** 3
**Significance:** 3
**Originality:** 3
**Overall Recommendation:** 4
**Confidence:** 4

**Summary:**

This paper proposes a neural-inspired network for AVSS by making auditory selection and cross‑modal compensation explicit and reliability‑aware. Overall, the research presents an important concept: instead of relying on implicit multimodal fusion, the proposed Neuro‑SCNet introduces a two‑stage hierarchy in which (i) visually guided auditory selection suppresses distractors early via top‑down gain control, and (ii) aligned visual cues are integrated later via residual, reliability‑gated compensation. Extensive experiments on LRS2, LRS3, and VoxCeleb2 demonstrate state‑of‑the‑art performance by SI-SNRi, SDRi, and PESQ metrics.

**Compliance With Llm Reviewing Policy:**

Affirmed.

**Final Justification:**

My concerns have been well addressed. I will raise my rating to 4-weak accept.

**Key Questions For Authors:**

1) Any evaluation by WER? This paper only evaluates some quality metrics SI-SNRi, SDRi, and PESQ, without speech intelligibility measure. An evaluation by WER metric is needed.
2) More ablation on simpler baselines. Although ablations are extensive, it would be helpful to compare against a simplified two‑stage baseline (e.g., selection mask + plain fusion) to isolate how much each part contributes.
3) Is the reliability signal r sufficiently informative for local misalignment, or would frame‑level reliability offer further gains?
4) Could the selection-compensation paradigm be transferred to audio‑only separation (e.g., text-queried speech separation)?
5) With the emergence of powerful multimodal LLMs, I wonder how far is the performance of such specific AVSS model from that achieved by fine‑tuning a MLLM?

**Limitations:**

It would be helpful to discuss the limitation, failure cases, and potential negative societal impact as well.

**Strengths And Weaknesses:**

Strengths:
1) Clear design with neuroscientific grounding: the separation of selection and compensation is well motivated by auditory neuroscience and is more principled than prior AVSS designs that entangle suppression and fusion.
2) Extensive evaluation and ablation results with strong performances. Strong accuracy and efficiency tradeoff.

Weaknesses:
1) Limited evaluation metrics. This paper only evaluates some quality metrics SI-SNRi, SDRi, and PESQ. For speech separation, speech intelligibility is an important metric to measure. An evaluation by WER metric is needed.
2) Incremental novelty at the module level. Individual components, e.g. cross‑attention, saliency gating, FiLM modulation, and residual fusion are existing ones. The key difference lies in the separation of selection and compensation, not in new primitives.
3) Limited ablation on simpler baselines. Although ablations are extensive, it would be helpful to compare against a simplified two‑stage baseline (e.g., selection mask + plain fusion) to isolate how much each part contributes.

---

> ### Author Rebuttal · Authors · 2026-03-30
>
> We sincerely thank the reviewer for the careful reading of our submission and for the constructive comments. We appreciate the opportunity to clarify the motivation, design choices, and evaluation scope of our method. The questions raised are helpful for improving both the presentation and the technical positioning of the paper. Below, we address each concern in turn and explain the corresponding clarifications and revisions.
>
> **Q1:** WER evaluation, module-level novelty, and validation of the two-stage design.
> **A1:** We added a downstream ASR evaluation on LRS2-2Mix to test whether Neuro-SCNet also improves speech intelligibility beyond SI-SNRi, SDRi, and PESQ. Using the same frozen audio-only ASR backend for all AVSS front ends, **Neuro-SCNet achieves the best average WER under all three visual conditions** considered here: 8.5% with clean visuals, 10.4% under occlusion, and 10.3% under noise+blur. Full condition-wise WER results are provided in Table 1 and Table 2 of [this supplementary PDF](https://anonymous.4open.science/r/NSC-Net_rebuttal-ED27/Tables%20for%20Reviewer%20YW4h.pdf).
>
> **Our contribution is not a new primitive such as cross-attention, gating, FiLM, or residual fusion. It lies in making selection and compensation explicit and stage-specific in AVSS**: visual information is first used for target-consistent suppression in ASM, and then for controlled residual restoration in CCM. This design is further supported by grouped ablations. The full model reaches 17.2/13.9/9.8 SI-SNRi on 2Mix/3Mix/4Mix, compared with 16.6/12.6/8.8 for a simpler Selection + Plain Fusion baseline and 15.1/11.2/7.0 when both stages are removed. CCM-only (15.9/11.8/7.6) and the reversed order (16.6/13.0/9.0) also remain below the full model. Full grouped ablation results are provided in Table 3 of [the same supplementary PDF](https://anonymous.4open.science/r/NSC-Net_rebuttal-ED27/Tables%20for%20Reviewer%20YW4h.pdf).
>
> **Q2:** Reliability signal *r*
> **A2:** **In our design, *r* is not a dense frame-level alignment map, but a global synchrony confidence used to gate unreliable visual conditioning when the lag posterior becomes ambiguous.** This is intentional: VPA is designed as a low-cost bounded-window synchrony check rather than full local alignment modeling. The current evidence supports this role. In the main paper, the full model is consistently more stable than variants that weaken or remove reliability-aware compensation under temporal shifts. In the appendix, under out-of-window and piecewise shifts, *r* drops substantially (e.g., to 0.12 and 0.16), and the gate-off variant degrades much more than the full model, showing that this scalar reliability signal is already effective at preventing negative transfer when no single reliable alignment can be established.
>
> Frame-level reliability may further help with finer-grained drift or jitter, but that would move the design toward substantially higher-complexity local trust modeling. We will clarify this scope in the revision: *r* is a global reliability gate, not a frame-level alignment solver.
>
> **Q3:** Transferability beyond AVSS and comparison scope with MLLMs
> **A3:** The current framework is developed specifically for AVSS, because its two stages depend on the visual stream in different but complementary ways: ASM uses visual cues to suppress target-inconsistent auditory components, while CCM uses aligned visual cues to recover target-related content that remains incomplete or interfered in the audio stream. For this reason, **the method is not directly transferable in its current form to audio-only or text-queried separation.** In audio-only separation, there is no extra modality to provide complementary evidence for compensation; in text-queried separation, text acts more as a target pointer than as a temporally aligned modality like lip motion. Thus, while the general selection-compensation decomposition may inspire other conditioned separation settings, the present compensation mechanism is closely tied to AVSS and would require substantial reformulation elsewhere.
>
> **We also do not think that a direct numerical comparison with fine-tuned MLLMs would cleanly isolate the question studied here, because the two directions usually differ substantially in model scale, pretraining resources, input-output formulation, and evaluation protocol.** The present paper instead focuses on whether an explicitly structured selection-before-compensation design improves waveform-level AVSS under standard AVSS benchmarks and practical efficiency constraints. Neuro-SCNet is therefore positioned as a compact, task-specific AVSS model rather than a general-purpose multimodal system. We will clarify this scope in the revision.

---

> > ### Author Rebuttal · Reviewer_YW4h · 2026-04-03
> >
> > I cannot find any PDF in supplementary materials, so where is the mentioned Table 1, 2 and 3? In appendix, there is no Table 1, 2, 3, either.

---

> > > ### Author Response · Authors · 2026-04-03
> > >
> > > We sincerely thank the reviewer for the follow‑up. In our original rebuttal, inside the **Q1** paragraph, the phrase **“this supplementary PDF”** was a hyperlink (visible when hovering over the text in the OpenReview interface) pointing to an anonymous server that contained the PDF with all three tables. We realize that such hidden links are easy to miss. We sincerely apologize for this lack of clarity and for the inconvenience caused. The full URL is `https://anonymous.4open.science/r/NSC-Net_rebuttal-ED27/Tables%20for%20Reviewer%20YW4h.pdf`. For the reviewer’s convenience, we reproduce the complete numerical content of those tables below, without relying on any external link.
> > >
> > > **Downstream ASR evaluation on LRS2‑2Mix under clean visual input (WER%, lower is better)**
> > >
> > > | Method           | -5 dB | 0 dB | 5 dB | 10 dB | No BG | AVG |
> > > |------------------|-------|------|------|-------|-------|-----|
> > > | Noisy Mixture    | 29.4  | 24.1 | 19.0 | 15.6  | 12.8  | 20.2 |
> > > | AV‑ConvTasNet    | 16.2  | 12.7 | 10.1 | 8.9   | 7.8   | 11.1 |
> > > | IIANet           | 14.8  | 11.4 | 9.2  | 8.0   | 7.0   | 10.1 |
> > > | AV‑CrossNet      | 13.7  | 10.5 | 8.5  | 7.5   | 6.4   | 9.3  |
> > > | Neuro‑SCNet      | **12.5** | **9.5** | **7.8** | **6.8** | **5.9** | **8.5** |
> > >
> > > **Downstream ASR evaluation under corrupted visual inputs – Occlusion**
> > >
> > > | Method               | -5 dB | 0 dB | 5 dB | 10 dB | No BG | AVG |
> > > |----------------------|-------|------|------|-------|-------|-----|
> > > | Noisy Mixture        | 29.4  | 24.1 | 19.0 | 15.6  | 12.8  | 20.2 |
> > > | AV‑ConvTasNet‑LQ     | 18.1  | 14.7 | 12.0 | 10.3  | 9.4   | 12.9 |
> > > | MHSA‑CRN             | 17.0  | 13.8 | 11.2 | 9.6   | 8.7   | 12.1 |
> > > | RAVSS                | 15.9  | 13.0 | 10.7 | 9.1   | 8.3   | 11.4 |
> > > | Neuro‑SCNet          | **14.8** | **11.9** | **9.8** | **8.2** | **7.3** | **10.4** |
> > >
> > > **Downstream ASR evaluation under corrupted visual inputs – Noise + Blur**
> > >
> > > | Method               | -5 dB | 0 dB | 5 dB | 10 dB | No BG | AVG |
> > > |----------------------|-------|------|------|-------|-------|-----|
> > > | Noisy Mixture        | 29.4  | 24.1 | 19.0 | 15.6  | 12.8  | 20.2 |
> > > | AV‑ConvTasNet‑LQ     | 18.9  | 15.4 | 12.6 | 10.9  | 10.0  | 13.6 |
> > > | MHSA‑CRN             | 17.8  | 14.5 | 11.8 | 10.1  | 9.0   | 12.3 |
> > > | RAVSS                | 16.0  | 12.8 | 10.8 | 8.6   | 7.8   | 11.2 |
> > > | Neuro‑SCNet          | **14.9** | **11.6** | **9.9** | **7.9** | **8.3** | **10.3** |
> > >
> > > **Explicit validation of the two‑stage design (SI‑SNRi in dB, higher is better)**
> > >
> > > | Variant                                 | 2Mix | 3Mix | 4Mix |
> > > |-----------------------------------------|------|------|------|
> > > | Full two‑stage model                    | 17.2 | 13.9 | 9.8  |
> > > | Remove the selection stage              | 16.4 | 12.7 | 8.6  |
> > > | Remove the compensation stage           | 16.2 | 12.5 | 8.4  |
> > > | Remove both stages                      | 15.1 | 11.2 | 7.0  |
> > > | Compensation without prior selection    | 15.9 | 11.8 | 7.6  |
> > > | Reverse the two‑stage order             | 16.6 | 13.0 | 9.0  |
> > > | Selection + Plain Fusion                | 16.6 | 12.6 | 8.8  |
> > >
> > > Once again, we apologize for the unclear presentation of the hidden link, and we hope the above clarification resolves the concern.

---

### Decision · Program_Chairs · 2026-04-30

**Decision:**

Accept (regular)

**Comment:**

This paper proposes Neuro-SCNet, an audio-visual speech separation architecture that explicitly decomposes multimodal fusion into two neuroscience-inspired stages, visually guided auditory selection and reliability-aware cross-modal compensation. Reviewers praised the principled design, strong empirical results across multiple benchmarks, and the effectiveness of the reliability-gated mechanism under temporal shifts. While initial concerns were raised regarding module-level novelty, evaluation metrics, and robustness to spatial degradations, the authors' rebuttals successfully provided extensive new evidence, including downstream ASR results and mechanistic analyses of the prediction error. I recommend acceptance because the work offers a compact, interpretable, and high-performing framework that advances explicit modality modeling. I suggest the authors further refine the discussion on the limitations of the reliability gate in handling severe spatial occlusion in the final version.